# Plant Extracts for Type 2 Diabetes: From Traditional Medicine to Modern Drug Discovery

**DOI:** 10.3390/antiox10010081

**Published:** 2021-01-09

**Authors:** Jinjoo Lee, Seungjin Noh, Suhyun Lim, Bonglee Kim

**Affiliations:** 1College of Korean Medicine, Kyung Hee University, Seoul 02447, Korea; leejinjoo1202@khu.ac.kr (J.L.); nohapril@khu.ac.kr (S.N.); lawinno@khu.ac.kr (S.L.); 2Department of Pathology, College of Korean Medicine, Kyung Hee University, Seoul 02447, Korea; 3Korean Medicine-Based Drug Repositioning Cancer Research Center, College of Korean Medicine, Kyung Hee University, Seoul 02447, Korea

**Keywords:** diabetes mellitus, plant extracts, antioxidant, glucose transport, inflammation, lipid metabolism

## Abstract

Type 2 diabetes mellitus (T2DM) is one of the largest public health problems worldwide. Insulin resistance-related metabolic dysfunction and chronic hyperglycemia result in devastating complications and poor prognosis. Even though there are many conventional drugs such as metformin (MET), Thiazolidinediones (TZDs), sulfonylureas (SUF), dipeptidyl peptidase 4 (DPP-4) inhibitors, glucagon like peptide 1 (GLP-1) and sodium-glucose cotransporter-2 (SGLT-2) inhibitors, side effects still exist. As numerous plant extracts with antidiabetic effects have been widely reported, they have the potential to be a great therapeutic agent for type 2 diabetes with less side effects. In this study, sixty-five recent studies regarding plant extracts that alleviate type 2 diabetes were reviewed. Plant extracts regulated blood glucose through the phosphoinositide 3-kinase (PI3K)/protein kinase B (Akt) pathway. The anti-inflammatory and antioxidant properties of plant extracts suppressed c-Jun amino terminal kinase (JNK) and nuclear factor kappa B (NF-κB) pathways, which induce insulin resistance. Lipogenesis and fatty acid oxidation, which are also associated with insulin resistance, are regulated by AMP-activated protein kinase (AMPK) activation. This review focuses on discovering plant extracts that alleviate type 2 diabetes and exploring its therapeutic mechanisms.

## 1. Diabetes Mellitus

Diabetes mellitus (DM) is the collective term for heterogeneous metabolic disorders whose main finding is chronic hyperglycemia. The cause is either a disturbed insulin secretion or a disturbed insulin effect, or usually both [1]. There are two primary forms of diabetes. Type 1 diabetes mellitus (T1DM) is caused by an absolute deficiency of insulin secretion. This form of diabetes results from autoimmune destruction of the β-cells of the pancreas or may occur idiopathically. However, type 2 diabetes mellitus (T2DM), which accounts for 90% to 95% of all diabetic patients, is referred to as non-insulin-dependent diabetes. It is identified by the combination of resistance to insulin action and an inadequate compensation for insulin secretion. Patients with T2DM may have insulin levels that appear normal or elevated. However, insulin secretion is insufficient for patients with higher glucose levels to compensate with insulin resistance [2]. There are several risk factors contributing to T2DM. First, genetic components act as a main factor. Higher concordance rates among twins and relatives, and susceptibility of loci discovered by genome-wide association studies ascribe T2DM to genetic correlation. The gut metagenome also contributes to the development of T2DM. Gut microbiota play a role in responding and interacting to the environment. T2DM patients tend to have a more hostile gut environment and a moderate degree of microbial dysbiosis. Acquired factors also have a great influence on T2DM. Obesity, sedentary lifestyle, physical inactivity, high-glycemic and low-fiber diet, vitamin deficiency, smoking and alcohol consumption act in complexity to induce T2DM [3]. T2DM is associated with numerous complications. Hyperglycemia, the common characteristic of T2DM, has the potential to cause devastating complications due to its insidious and chronic nature [4]. The chronic elevation of blood glucose levels damages blood vessels, leading to vascular complications. Microvascular complications include retinopathy, nephropathy, and neuropathy. Macrovascular complications include cardiovascular disease manifesting as myocardial infarction and cerebrovascular disease such as strokes [5]. Diabetic gastroenteropathy is also one of the common complications in prolonged diabetic patients. It causes various symptoms such as heartburn, abdominal pain, nausea, vomiting, constipation, diarrhea, and fecal incontinence [6]. Diabetic complications are considered as serious problems regarding patients’ impaired quality of life, poor prognosis, and high morbidity.

## 2. Conventional Treatments for Diabetes Mellitus

Currently, there are many pharmacological treatment options for diabetic patients. Metformin (MET) has been widely used as a first-line approach for T2DM patients due to its durable anti-hyperglycemic effects, low risk of hypoglycemia, robust cardiovascular safety profile, and low cost. However, 25% of patients cannot tolerate the medication in adequate amounts due to metformin-associated gastrointestinal side effects. Metformin may have direct serotonergic-like effect or alter the transport of serotonin, which is associated with nausea, vomiting and diarrhea [7]. Thiazolidinediones (TZDs) decrease insulin resistance directly through activation of peroxisome proliferator-activated receptor γ (PPARγ), which has primary effects on adipose tissue and decreases insulin resistance. However, the complications of TZD therapy include increased risk of heart failure, bone fractures, weight gain, fluid retention, and edema [8]. Sulfonylureas (SUF) bind the ATP-dependent K+ channels on beta-cells membrane, therefore inducing the depolarization of the plasma membrane. Consequently, voltage-gated Ca^2+^ channels are opened and the spike of intracellular Ca^2+^ levels triggers insulin secretion. Even though they can effectively lower blood glucose and increase insulin secretion, their highest glucose-lowering potential inevitably involves risk of severe hypoglycemia especially in elderly people with impaired kidney function. Additionally, SUF usually lead to moderate weigh gain. Dipeptidyl peptidase 4 (DPP-4) inhibitors are the replaceable therapy with sulfonylureas. DPP-4 inhibitors show a safety profile in patients with renal insufficiency, similar anti-hyperglycemic effects with low hypoglycemic rates, and weight-neutral effects. However, an increased incidence of acute pancreatitis and inflammatory bowel disease after therapy with DPP-4 inhibitors were observed in some studies. Glucagon-like peptide 1 (GLP-1) receptor agonists are antidiabetic drugs applied subcutaneously. They have greater plasma glucose-lowering effects than oral antidiabetics and have blood pressure-lowering, weight-reducing, and cardio- and renal-protective effects. However, transient nausea, vomiting, and diarrhea are adverse effects induced by GLP-1 receptor agonists [9,10]. Sodium-glucose cotransporter-2 (SGLT-2) inhibitors were proven to be effective in glycemia and HbA1c reduction by lowering glucose renal reabsorption at the proximal tubules in the kidney. SGLT2 inhibitors have been recommended by clinical guidelines as potential pharmacological approaches for second-line therapy following metformin failure or intolerance [11]. However, significantly increased risks of genital infections with SGLT-2 inhibitors were observed in some studies [9]. 

Although various antidiabetic drugs have been developed, the associated side effects, such as gastrointestinal symptoms, heart failure, weight gain, edema, impaired kidney function pancreatitis, and genital infections, etc., became another burden on patients. Treatments with less side effects are necessary, and plant extracts might be an effective therapeutic intervention. Therefore, this study aims to review plant extracts with antidiabetic efficacy based on the pathogenesis of T2DM. 

## 3. Pathogenesis of Type 2 Diabetes

### 3.1. Glucose Transport and Metabolism

The phosphoinositide 3-kinase (PI3K)/protein kinase B (Akt) pathway and Adenosine 5′-monophosphate (AMP)-activated protein kinase (AMPK) play a central role in glucose homeostasis. When insulin binds to the extracellular α-subunits of the insulin receptor, insulin signaling is initiated. This interaction induces conformational changes and facilitates tyrosine phosphorylation on the β-subunit of the insulin receptor. Then, insulin receptor substrates (IRS) are tyrosine phosphorylated, and bind with PI3K. Activated PI3K converts phosphatidylinositol-4,5-bisphosphate (PIP2) to phosphatidylinositol-3,4,5-triphosphate (PIP3) at the plasma membrane, which facilitates the activation of Akt. Activated Akt translocate to numerus cellular compartments to regulate downstream effectors [12]. First, Akt induces the translocation of glucose transporter 4 (GLUT4), which is translocated to the plasma membrane and transports glucose into skeletal muscle. Second, Akt converts glucose to glucose 6-phosphate (G6P) by stimulating hexokinase, which produces cellular energy via glycolysis. Third, activated Akt promotes glycogen synthesis in skeletal muscle. Akt inhibits glycogen synthase kinase 3 (GSK3), which exerts an inhibitory effect on glycogen synthase (GS) [13]. AMPK is a major regulator of energy metabolism, especially in diabetes and metabolic diseases. AMPK is a heterotrimeric enzyme comprised of a catalytic (α1 or α2) subunit and two regulatory (β1 or β2 and γ1, γ2, or γ3) subunits. AMPK controls glucose homeostasis mainly through the inhibition of gluconeogenic gene expression and hepatic glucose production [14]. Gluconeogenesis, the increase in hepatic glucose production, is a vital element in the progress of glucose disorders. If liver glycogen synthesis and gluconeogenesis remain in a disequilibrium, elevation of blood glucose appears [15]. AMPK activation suppresses gluconeogenesis by inhibiting two key gluconeogenic enzymes, which are phosphoenolpyruvate carboxykinase (PEPCK) and glucose-6-phosphatase (G6Pase) [16]. In addition, AMPK activation increases the phosphorylation of GSK-3β and thereby reduces PEPCK gene expression in the liver, reducing gluconeogenesis [14]. The PI3K/Akt pathway and AMPK play an important role in regulating glucose metabolism and energy homeostasis. PI3K/Akt-mediated glucose metabolism or AMPK activation dysfunction lead to T2DM. Therefore, using the PI3K/Akt pathway and AMPK as therapeutic targets might be an effective antidiabetic intervention. Mechanisms of plant extracts regulating glucose transport and metabolism are organized in Figure 1.

### 3.2. Inflammation and Oxidative Stress

Inflammation and oxidative stress are directly associated with insulin resistance through activating the c-Jun amino terminal kinase (JNK) and IκB kinase-β (IKKβ)/nuclear factor kappa B (NF-κB) pathways. JNK has been shown to promote insulin resistance by inhibiting the signal through insulin receptor/IRS-1 axis. IRS-1 is a substrate of the insulin receptor that undergoes phosphorylation in response to insulin. Tyrosine-phosphorylated IRS proteins leads to the activation of downstream effectors of normal insulin signaling. However, serine/threonine phosphorylation negatively modulates insulin signaling by hindering tyrosine phosphorylation sites, inducing IRS protein degradation and dissociation from insulin receptors [17,18]. JNK induces the phosphorylation of IRS-1 at serine sites, and therefore disrupts normal signaling through the insulin receptor/IRS-1 axis. The IKKβ/NF-κB pathways causes insulin resistance through transcriptional activation of NF-κB. The activation of IKKβ targets IκBα for proteasomal degradation, leading NF-κB to translocate into the nucleus. Then, the expression of numerous target genes producing inflammation mediators, such as tumor necrosis factor- α (TNF-α), interleukin-1β (IL-1β), interleukin-6 (IL-6), monocyte chemotactic protein 1 (MCP-1), toll-like Receptor (TLR), cyclooxygenase-2 (COX2), C-reactive protein (CRP), etc., are increased, which then cause insulin resistance. Thus, the activation of the JNK and NF-κB pathways promotes the development of insulin resistance and T2DM. Pro-inflammatory cytokines and chemokines including TNF-α, IL-6 and IL-1β activate both the JNK and NF-κB pathways. Reactive oxygen species (ROS) and endoplasmic reticulum (ER) stress also activate the JNK and NF-κB pathway [17]. In conclusion, inflammation and oxidative stress activate inflammatory signaling pathways which directly or indirectly block insulin action. Targeting inflammatory mediators and antioxidant enzymes can be another approach for antidiabetic therapy. Mechanisms of plant extracts with anti-inflammatory and anti-oxidative properties are organized in Figure 2.

### 3.3. Lipid Metabolism

AMPK is also recognized as a key regulator of lipid metabolism. AMPK interacts with and directly phosphorylates sterol regulatory element binding proteins (SREBP). SREBP is a key lipogenic transcription factor that is regulated by glucose and insulin. SREBP-1c regulates the lipogenic process related to fatty acid and triglyceride synthesis, whereas SREBP-2 controls cholesterol synthesis and uptake. Thus, the inhibition of AMPK and the activation of SREBP-dependent lipogenesis contribute to the development of insulin resistance [19]. AMPK also decreases hepatic lipogenesis by directly phosphorylating acetyl-CoA carboxylase (ACC). ACC is a rate-determining enzyme for the synthesis of malonyl-CoA, critical for fatty acid biosynthesis and the inhibition of fatty acid oxidation. The phosphorylation and inactivation of ACC1 leads to the inhibition of fatty acid and cholesterol synthesis. Phosphorylation of ACC2 increases fatty acid oxidation [14]. Simultaneously, AMPK also promotes fatty acid oxidation by relieving the suppression of carnitine palmitoyltransferase1 (CPT1) [20]. The dysregulation of AMPK and downstream effectors demonstrates the pathogenesis of hepatic steatosis, dyslipidemia, and insulin resistance. The broad activity of AMPK in lipid metabolism makes it a very potential therapeutic target with T2DM and related metabolic disorders. Mechanisms of plant extracts regulating lipid metabolism are organized in Figure 3.

## 4. Type 2 Diabetes and Plant Extracts

The association of plant extracts with type 2 diabetes mellitus (T2DM) has been shown through their therapeutic effects [21]. The efficacy can be demonstrated by laboratory results and four major mechanisms, which include glucose transport and metabolism, anti-inflammation and antioxidant activity, lipid metabolism, etc. Several studies have been reported to include various experiment models such as in vitro, in vivo, and clinical trials.

### 4.1. In Vitro Studies

Several in vitro studies reported antidiabetic effects of plant extracts (Table 1). Various cell lines are manipulated to discover the antidiabetic effects of plant extracts. 3T3-Ll preadipocytes are cloned from 3T3 mouse embryo fibroblasts, and their differentiation is associated with lipid-genesis as well as the activity of lipid related enzymes [22]. Chinese hamster ovary (CHO)-K1 cell lines are one of the most preferred host cells for the manufacturing of complex therapeutics due to high human compatibility [23]. Rat L6 muscle cells are capable of mimicking the characteristics of mature muscle cells and are unique in expressing GLUT4 transporters in response to insulin [24]. Rin-5f cell lines are insulin-secreting pancreatic cancer cells and are used to confirm various insulin- and diabetes-related mechanisms [25]. Hep G2 cell lines are derived from human hepatoblastoma and are used to express numerous liver-specific metabolic functions, including metabolisms of lipoprotein, cholesterol and insulin [26]. In addition, immortalized primary human hepatocytes (HuS-E/2) H4IIE hepatoma cells, rat insulinoma cell line (INS-1), and skeletal muscle cells from various origins were used to confirm the efficacy of plant extracts. Han et al. demonstrated the efficacy of *Anemarrhena asphadeloides Bge. extract on* 3T3-L1 and LKB1-deficient HeLa cells with the *administration of* 30 μg/mL concentration for 2 h [27]. The phosphorylation of AMPK and ACC was activated in both cell types, indicating a LKB1-independent antidiabetic mechanism of the extract. Haselgrübler et al. suggested *Bellis perennis* extract as an effective inducer of glucose transporter 4 (GLUT4) translocation in the absence of insulin [28]. Starved CHO-K1 cells expressing the human insulin receptor and GLUT4-myc-GFP fusion protein were incubated with the extract at a concentration of 1 mg/L for 10 min. The efficacy of *Bellis perennis* extract in reducing blood glucose levels in a living organism (in ovo) was confirmed. Zhao et al. demonstrated that Folium Sennae ethanolic extract, derived from *Cassia angustifolia Vahl*, promoted glucose uptake by stimulating GLUT4 expression and translocation in L6 cells [29]. GLUT4 translocation was promoted via several signaling pathways, including the AMP-activated protein kinase (AMPK), phosphoinositide 3-kinase (PI3K)/protein kinase B (Akt), protein kinase C (PKC), and G protein–phospholipase C (PLC)–PKC signaling pathways. In particular, the G protein–PLC–PKC signaling pathway and inositol 1,4,5-trisphosphate receptor (IP3R) triggered the increase in intracellular Ca^2+^ release, which plays a crucial role in the glucose uptake process. Bowser et al. reported that cocoa (*Theobroma cacao*) extract elicited antidiabetic effects by mediating glucose homeostasis [30]. Human primary skeletal muscle cells treated with 10, 25 μM for 2 h displayed improvements in basal glucose uptake, following the increase in basal glycogen synthesis and insulin-induced glycogen synthesis. These results were not related with the AMPK of calcium/calmodulin-dependent protein kinase II (CaMKII) activation, and it was further discussed that single-compound procyanidins from cocoa showed better antidiabetic effects. *Coptischinensis franch* acid extract (CCE) indicated the stimulation of pancreatic insulin secretion, acting as a treatment for T2DM [31]. The Rin-5f cell line treated with 2, 10, 50, 100, 250 or 500 µM concentrations for 24 h exhibited increased glucose-stimulated insulin secretion (GSIS) and diminished insulin secretion. CCE also exerted a protective role on islet β-cells by increasing islet β-cell proliferation and the protein expression of PARP-1, thus enhancing insulin sensitivity. Song et al. demonstrated that an aqueous extract of *Dendropanax morbifera* (DLW) inhibited adipogenesis by increasing activities of CCAAT/Enhancer Binding Protein (CEBP) α, CEBPβ, peroxisome proliferator-activated receptor γ (PPARγ), and sterol response element binding protein 1 (SREBP1) [32]. Mice 3T3-L1 adipocytes were treated with 50, 100, 300 and 500 μg/mL of DLW extracts for 7 days. Decreased levels of serum insulin, serum leptin and serum adiponectin were observed. In addition, the mRNA and protein expression levels of adipogenesis-related genes were significantly lowered by DLW. Hetta et al. estimated the antidiabetic effects of *Eruca sativa* (rocket salad) leaf extract upon major insulin-responsive cells, including C2C12 skeletal muscle cells, H4IIE hepatocytes and 3T3-L1 adipocytes [33]. An N-haxane-soluble fraction of 95% ethanol extract showed the highest efficacy among various types of ethanol fractions. The C2C12 skeletal muscle myoblast showed an increase in glucose uptake when treated with 12.5 μg/mL for 18 h, while H4IIE hepatoma cells showed a decrease in glucose-6-phosphatase (G6Pase) activity with the same concentration for 16 h. An elevation of intracellular triglycerides (TG) content was detected in 3T3-L1 adipocytes with a concentration of 6.25 or 12.5 μg/mL for 8 h. Chang et al. reported that *Helminthostachys zeylanica* extract exhibited antidiabetic properties by inhibiting fatty acid synthesis and activating fatty acid β-oxidation [34]. Administration of a concentration of 100 μg/mL, for 18 h, of *palmitate-treated* HuS-E/2 cells showed increased levels of p-AMPK, p-ACC, CPT1, PPARα and PPARδ, and suppressed SREBP-1c and *PPARγ* activity. Gao et al. demonstrated that Sea Buckthorn fruit oil extract showed promising effects on the PI3K/Akt pathway dependent insulin resistance, which originated from *Hippophae rhamnoides* L., traditionally used for sputum, coughs, skin diseases, and dyspepsia [35]. Insulin-resistant HepG2 cells were treated with 400 μM for 24 h and showed a significant rise in glucose uptake, GS, PI3K, and p-Akt, as well as a reduction in GSK-3β expression. Park et al. illustrated that an ethanol extract of *Mori ramulus* refined β-cell dysfunction and insulin resistance by reducing oxidative damage and advanced glycation end-product formation [36]. INS-1 cells were administered with 62.5, 125, 250, 500, and 1000 μg/mL for 1 h, and exhibited anti-glycation effects and improvements of PDX-1. Yan et al. elicited the regulatory effects of *Morus alba* L. anthocyanin extract against insulin resistance by the stimulation of the PI3K/Akt pathway [37]. Insulin-resistant HepG2 cells, treated with 50, 100, and 250 μg/mL concentrations for 24 h, showed an increase in glucose consumption which was supported by elevated levels of glycogen. The lowered enzyme activities of G6Pase and PEPCK were due to the regulation of PGC-1α and the phosphorylation of forkhead Box O1 (FOXO1). The phosphorylation of Akt and GSK3β also led to an upregulation of glycogen synthase 2 (GYS2). Vlavcheski et al. reported the potential of *Rosmarinus officinalis* L. extract to counteract insulin resistance in palmitate-induced L6 muscle cells [38]. Exposure of the cells to palmitate resulted in the elevation of insulin receptor substrate 1 (IRS-1), c-Jun N-terminal kinase (JNK), mammalian target of rapamycin (mTOR) and p70 S6K phosphorylation, and reductions in insulin-stimulated Akt phosphorylation and glucose uptake. These palmitate-induced responses were diminished by the administration of *Rosmarinus officinalis* L. extract at a concentration of 5 μg/mL for 16 h. Increased phosphorylation of AMPK was also observed even in the presence of palmitate.

A total of twelve studies demonstrated the efficacy of plant extracts against diabetes. Seven plant extracts were found to be effective in regulating glucose transport and metabolism. Especially, *Cassia angustifolia* Vahl ethanolic extract and *Rosmarinus officinalis* L. extract increased the level of GLUT4, which was regulated by increased phosphorylation of Akt and AMPK [29,38]. *Dendropanax morbifera* water extract and *Helminthostachys zeylanica* extract were effective in normalizing lipid genesis through the downregulation of SREBP-1c, C/EBPα and C/EBPβ, which are known to be related with the differentiation of preadipocytes [32,34,39]. Meanwhile, *Coptischinensis franch* acid extract and *Dendropanax morbifera* water extract showed anti-inflammatory activities by the regulation of PARP-1 and FAS, respectively [31]. These findings all together supported the antidiabetic properties of the plant extracts. 

### 4.2. In Vivo Studies

Numerous in vivo studies have demonstrated the efficacy of plant extracts against diabetes. Studies have investigated whether the extracts effectively alleviated pathological conditions in animal models of diabetes. The most commonly used method to induce a diabetic model was using Streptozotocin (STZ) intraperitoneal or intravenous injection [40]. STZ exerts selective cytotoxic activity upon insulin-producing β-cells in the pancreas, which inducing both type 1 and type 2 diabetes [41]. According to a study that examined the reliability of STZ injection as a method to induce diabetes, hyperglycemia and hyperlipidemia was immediately induced in Sprague Dawley (SD) rats after injection. Diabetic complications were detected in long-term injection, along with the increase in inflammatory cytokines, and the damage of pancreatic cells in histological examination [42]. In addition, a high-fat and -fructose diet is another method utilized to induce diabetic models. Meta-inflammation due to persistent high-fat diet affects the signaling pathway of insulin receptors, which consequently interferes with insulin-mediated blood glucose regulation [43]. Furthermore, a high-fructose diet resulted in lipid abnormalities, impaired glucose tolerance, and increased oxidative stress [44]. SD rats, Wistar rats, C57BL mice, KK-Ay mice were used as study models to induce diabetes. Other diabetic models including Tsumura Suzuki Obese Diabetes (TSOD) mouse and Institute of Cancer Research (ICR) mouse were also utilized. These diabetic models showed improvements in diabetic symptoms after treatment with plant extracts, and revealed relevant mechanisms according to glucose and lipid metabolism as well as inflammatory responses. 

#### 4.2.1. Sprague Dawley (SD) Rats

Sprague Dawley (SD) rats are outbreds from Wistar rats, which are produced by the Charles River Laboratories (CLR) and Harlan Laboratories (HAR). SD rats are extensively used for the development of animal models, particularly related to metabolic disorders [45]. In particular, STZ-induced models are utilized to mimic various diseases including cardiovascular diseases and diabetes, while other methods are also widely used to induce infection or cancer models [46,47,48,49]. Several studies were conducted on SD rats in order to identify the efficacy of plant extracts against diabetes (Table 2). Mohammed et al. reported that the ethyl acetate fraction of Aframomum melegueta K. Schum. ethanolic extract showed anti-type 2 diabetes (T2DM) activity via improving hyperglycemia, insulin sensitivity, and dyslipidemia [50]. Type 2 diabetes SD rats were orally treated with 150 or 300 mg/kg for 4 weeks, and the effects were more pronounced in the high-concentration group. Increased levels of insulin, homeostasis model assessment of β-cell function (HOMA-β) and high-density lipoprotein cholesterol (HDL-C), and decreased levels of α-amylase, α-glucosidase, non-fasting blood glucose (NFBG), fructosamine, homeostasis model assessment of insulin resistance (HOMA-IR), total cholesterol (TC), TG, low-density lipoprotein cholesterol (LDL-C), atherogenic index (AI), and coronary risk index (CRI) were observed. Diabetic complication-related parameters, such as alanine aminotransferases (ALT), aspartate aminotransferases (AST), alkaline phosphatase (ALP), urea, uric acid, creatinine, lactate dehydrogenase (LDH), creatine kinase (CK-MB) and diabetes-induced pancreatic damage, were also improved. Han et al. suggested that *Anemarrhena asphadeloides Bge.* extract had antidiabetic effects on Bacillus Calmette–Guérin (BCG) vaccine-induced insulin resistance SD rats [27]. After being treated with concentrations of 20, 60 and 180 mg/kg for 14 days, the glucose infusion rate (GIR) was elevated, suggesting enhanced insulin action. Jeong et al. examined the efficacy of *Codonopsis lanceolate* water extract (CLW) through feeding a high-fat diet containing 0.3, 1% w/w of CLW to SD rats for 8 weeks [51]. CLW downregulated serum insulin levels and the size of pancreatic β-cells. Hepatic insulin sensitivity increased, which was associated with enhanced insulin signaling from p-Akt to phosphorylated glycogen synthesis kinase-1β (pGSK-1β). Phosphoenolpyruvate carboxykinase (PEPCK) levels decreased, while carnitine palmitoyltransferase 1 (CPT-1), p-AMPK levels were increased. Consequently, CLW consumption led to improved insulin sensitivity and the regulation of insulin secretion capacity, effectively alleviating diabetic symptoms. *Coptischinensis Franch* acid extract (CCE) was indicated to modulate pancreatic insulin secretion, acting as a treatment for T2DM [31]. High-fat diet, STZ-induced T2DM in male SD rats were administered with CCE at a dose of 100 mg/kg for 8 weeks. Decreased body weight, fasting blood glucose and basal insulin levels were confirmed, which showed that CCE could relieve diabetes-related symptoms. Yang et al. examined the effects of *Gastrodia elata* Blume water extract against high-fat diet-induced diabetic male SD rats to treat type 2 diabetes [52]. Eight weeks of supplementation of 0.5% and 2% of the extract lowered serum glucose levels during oral glucose tolerance test (OGTT), and hepatic glucose output during hyperinsulinemic clamp state. It improved glucose infusion, glucose uptake in gastrocnemius and quadriceps muscles, and insulin sensitivity during hyperglycemic state. The mediation of carbohydrate and fat oxidation was also detected. Histological examinations showed increased number, size, and mass of β-cells, as well as a decrease in β-cell apoptosis. Gao et al. demonstrated that Sea Buckthorn fruit oil extract originating from *Hippophae rhamnoides* L. showed promising effects on PI3K/Akt pathway-dependent insulin resistance [35]. High-fat diet-induced type 2-diabetic male SD rats were administered with oral doses of 100, 200 or 300 mg/kg/day for 4 weeks, and this resulted in a decrease in insulin, blood glucose levels and ALT and AST, which was further explained by the increase in hepatic glycogen. Man et al. suggested that *Litchi chinensis* Sonn. seeds ethanol extract inhibited glycogenesis, proteolysis, and lipogenesis in high-fat diet- and STZ-induced T2D SD rats [53]. Decreased insulin resistance index, urinary sugar, serum ALT, serum AST and water consumption were observed. Glucose and fatty acid metabolisms were regulated via upregulating the expression of Glu2, Glu4, insulin receptor and IRS2. In the liver, impairments of IRS2, PI3K, Akt and mTOR insulin signaling were restored. Wu et al. claimed the improvement of cognitive impairment and neuronal injury in administrating *Litchi chinensis* Sonn. seed 70% ethanol extract in high-fat diet- and STZ-induced T2DM SD rats [54]. The levels of glucose, insulin, amyloid β (Aβ), AGEs and Tau protein were significantly increased in the blood and hippocampus of T2DM rats. Lychee seed extract showed similar effects to donepezil, a classic pharmaceutic for Alzheimer’s disease (AD), on the spatial learning and memory of the rats. Al-Zuaidy et al. reported that *Melicope lunu-ankenda* leaves extract exhibited antidiabetic properties [55]. High-fat diet- and STZ-induced diabetes SD rats were administered with 200 and 400 mg/kg of the extract for 8 weeks. The treatment increased the insulin sensitivity of obese rats, while reducing total triglyceride, cholesterol and LDL. However, cholesterol and HDL levels were significantly elevated when treated with the extract. Disturbances in glucose metabolism, tricarboxylic acid cycle, lipid metabolism, and amino acid metabolism were ameliorated. Ma et al. suggested that treatment of 70 mg/kg *Momordica charantia* L. with 70% ethanol (MCE) extract in high-fat diet- and STZ-induced diabetes SD rats for 6 weeks significantly improved insulin resistance [56]. Lower body weight, fasting serum glucose, fasting serum insulin and HOMA-IR levels were observed. Reductions in tumor necrosis factor- α (TNF-α), interleukin-6 (IL-6), GLUT-4, suppressor of cytokine signaling 3 (SOCS-3), JNK, and Akt expression were expressed. MCE inhibited the phosphorylation of JNK and nuclear translocation of nuclear factor-kB (NF-κB), which contributed to the improvements in insulin signaling and inflammation. These results confirm that MCE ameliorated insulin resistance (IR) by downregulating SOCS-3, JNK mRNA and protein expression. Ma et al. suggested that *Mori Cortex* 70% alcohol extract (MCE) administered at a dose of 10 g/kg for 12 weeks in high-fat diet-, STZ-induced diabetes SD rats downregulated blood lipid levels and reserved insulin resistance [57]. The expression levels of SREBP-1c and carbohydrate-responsive element-binding protein (ChREBP) were measured in liver samples of the rat models. MCE decreased the protein and mRNA expression levels of SREBP-1c and ChREBP, which indicates the protective effect of MCE on hepatic injury that commonly occurs with type 2 diabetes mellitus (T2DM). Cai et al. demonstrated that *Morus alba* L. leaf extract showed hypoglycemic and hypolipidemic effects against fructose induced diabetic male SD rats [58]. Treated with a daily dose of 2 g/kg for 4 weeks, it was observed that fasting blood glucose level, insulin resistance, TG, TC, and LDL decreased, and lipid accumulation in skeletal muscles was inhibited. The upregulation of IRS-1, PI3K, p85a, and GLUT4 in skeletal tissues suggested the potential of *Morus alba* L. leaf extract to activate the IRS-1/PI3K/Glut4 signaling pathway. Mousum et al. suggested that nephrotoxicity triggered by severe T2DM could be suppressed by the administration of *Nyctanthes arbor-tristis* L. leaf ethanol extract [59]. High-fat diet-fed and STZ-sensitized diabetes SD rats with 200 and 400 mg/kg of the extract for 4 weeks were monitored. The suppression of hyperglycemia-mediated oxidative stress and inflammatory cascades via controlling NF-kBp65 expressed dose-dependent hypoglycemic and hypolipidemic activity. The impaired architectures of kidney, aorta, and tissues were recovered after the treatment. Ibrahim et al. demonstrated that the butanol fraction of *Parkia biglobosa* (Jacq.) G. Don leaves stimulated insulin secretion and prevented complications due to type 2 diabetes [60]. STZ-induced T2DM male SD rats were treated with a dose of 150 mg/kg, 5 days a week for 4 weeks. β-cell function and insulin secretion improved, and insulin resistance and liver glycogen were restored. Other related factors such as fructosamine, ALP, and urea were downregulated, while histological examinations showed increases in the number of β-cells. Liu et al. suggested that *Phellinus Linteus* extract effectively dampened blood glucose fluctuation by reducing glycosylated serum protein (GSP) level, improved insulin resistance, and ameliorated liver and kidney injury [61]. It downregulated fructose-1,6-bisphosphatase (FBPase) and G6Pase, the key gluconeogenesis enzymes leading to hyperglycemia, and upregulated GLUT2 and glucokinase (GCK), enhancing glucose transfer from blood to liver and glycolysis. Additionally, the elevation of acyl-CoA oxidase 1 (ACOX1), carnitine palmitoyltransferase 1A (CPT1A) and low-density lipoprotein receptor (LDLR), and the inhibition of 3-hydroxy-3-methylglutaryl-CoA reductase (HMGCR) indicated the attenuation of hyperlipidemia. These were associated with lipid metabolism, including fatty acid β-oxidation and cholesterol synthesis. Cam et al. reported that an intake of *Thymus praecox* subsp. *skorpilii* var. *skorpilii* methanolic extract showed antidiabetic, hepato-protective, and anti-inflammatory effects by reducing glucose, ALT, creatinine (CR), TNF-α, IL-1β and IL-6 [62]. The suppression of sodium glucose co-transporters (SGLT)-1 and 2 and the elevation of hexokinase (HK) were associated with glucose absorption and utilization, respectively. Activation of *PPARγ* inhibited the activation of PEPCK, while consequently facilitating glucose balance. The number of Langerhans islets also increased after the treatment at a dose of 100 mg/kg for 3 weeks in an STZ/nicotinamide (NA)-induced T2DM SD rat model. Mohammed et al. showed that the acetone fraction of *Xylopiaaethiopica* (Dunal) A.Rich. fruit had antidiabetic effects against fructose diet-induced T2DM male SD rats [63]. The daily administration of 150 or 300 mg/kg for 4 weeks showed improvements in β-cell function and serum insulin. Decreases in insulin resistance, fructosamine, and artherogenic/cardiogenic index were observed. Lipid metabolism and liver function showed recovery through the regulation of liver glycogen, ALT, and CK-MB. Histopathological findings further revealed larger islets with higher numbers of β-cells. Saravanan et al. demonstrated that dietary ginger, originating from *Zingiber officinale* Roscoe, promoted the recovery of glucose dysregulation, lipotoxicity, and oxidative stress in high-fat, high-fructose diet-induced prediabetic male SD rats [64]. The provision of 3% ginger powder daily for 8 months improved glucose tolerance and disappearance, while it lowered insulin levels, insulin resistance and triglycerides. Histopathological findings detected insulin positivity in the pancreas, supporting antidiabetic effects of ginger. Ibrahim et al. elucidated that the butanol fraction of *Ziziphus mucronata* Willd ameliorated glucose dysregulation and dyslipidemia in STZ-induced diabetic Male SD rats [65]. Treated with 300 mg/kg of the extract for 4 weeks (5 days a week), glucose tolerance, serum insulin levels, and liver glycogen levels improved, while blood glucose levels were downregulated. However, other diabetic parameters such as HOMA-b, HOMA-IR, serum fructosamine levels, and hepatic and renal function tests were not significantly affected. These results all together supported the efficacy of plant extracts against glucose and lipid dysregulation, as well as various diabetic symptoms including hepatic, cardiac, and inflammatory disorders. 

Ninteen studies were conducted with diabetes-induced SD rats, and most of the extracts showed effectiveness in regulating insulin secretion or resistance. *Litchi chinensis* Sonn. was the only extract that was mentioned in more than one study. It improved glucose and lipid metabolism while relieving neuronal damage and cognitive disorders, which is one of the various diabetic complications [53,54]. *Codonopsis lanceolate* water extract, *Momordica charantia* L. 70% ethanol extract, and *Thymus praecox* subsp. skorpilii var. skorpilii methanolic extract affected all mechanisms of glucose metabolism, lipid metabolism, and inflammation [51,56,62]. Plant extracts that were particularly related with glucose metabolism included *Aframomum melegueta* K. Schum. fruit ethanolic extract ethyl acetate fraction*, Gastrodia elata* Blume water extract*, Litchi chinensis* Sonn. seeds ethanol extract, *Morus alba* L. leaf extract, and *Phellinus Linteus* mycelial extract [50,52,53,58,61]. The mechanisms that were mainly identified were the activation of GLUT-4, p-Akt and PI3K, while other enzymes such as α-glucosidase were also mentioned. Meanwhile, *Nyctanthes arbor-tristis* L. leaf ethanol extract and *Mori* Cortex 70% alcohol extract especially showed relevance in regulating inflammatory changes and lipid metabolism, respectively [57,59]. However, several extracts including *Melicope lunu-ankenda* leaf extract, *Nyctanthes arbor-tristis* L. leaf ethanol extract, *Phellinus Linteus* mycelial extract, and *Ziziphus mucronata* Willd ethanol extract were treated with doses higher than 200 mg/kg [55,59,61,65]. As these extracts were administered at excessively high concentrations, it was necessary to reconsider the effectiveness of these experiments.

#### 4.2.2. Wistar Rats

Wistar rats are standardized rodent models produced by the Wistar Institute, which are widely used to mimic various types of diseases [66]. Physiological changes of Wistar rats with age are similar to the pathological changes in humans suffering from metabolic disorders [67]. STZ- induced Wistar rats have been proven to effectively imitate symptoms caused by both type 1 and type 2 diabetes [68,69]. Numerous studies were conducted on diabetes-induced Wistar rats and these animal models showed significant recovery from diabetic symptoms after the treatment with plant extracts (Table 3). Hocayen et al. indicated that the administration of *Baccharis dracunculifolia DC. Asteraceae* extract at a dose of 400 mg/kg for 30 days promoted protective effects on monosodium glutamate (MSG)-induced pancreatic damage in obese Wistar rats [70]. Reductions in the area under the pancreatic islets were protected, and a higher concentration of insulin secreted by islets was observed. The highest antioxidant capacity was obtained with ethanol as the extracting solvent, followed by methanol and acetone. The ethanolic and methanolic plant extracts exhibited significant 1,1-diphenyl-2-picrilhidrazyl (DPPH) and 2,2-azinobis-(3-ethylbenzothiazoline-6-sulfonic acid) (ABTS+) free radical-scavenging ability and ferric-reducing power. Gomaa et al. reported that *Boswellia serrata* extract administration decreased the hippocampal levels of Aβ 1-42, p-tau, caspase-3, cholinesterase (ChE) and glycogen synthase kinase-3β (GSK-3β), and elevated glutamate receptors’ gene expression [71]. This effect is attributed to the downregulation of TNF-α, IL-1β, IL-6, malondialdehyde (MDA) and the upregulation of glutathione (GSH) and superoxide dismutase (SOD) levels in the hippocampus of the T2DM Wistar rat model. Reduced glucose, insulin, cholesterol and HOMA-IR were also observed after the treatment. This result suggested that *Boswellia serrata* extract ameliorated cognitive impairment and insulin resistance associated with type 2 diabetes. Bem et al. indicated that *Euterpe oleracea* Mart. hydroalcoholic extract decreased blood glucose, insulin resistance, leptin and IL-6 levels, lipid profile, and vascular dysfunction, while increasing the expression of insulin signaling proteins in skeletal muscle, adipose tissue and plasma glucagon like peptide-1 (GLP-1) levels [72]. Metabolic changes, such as increases in blood glucose, serum insulin, glycosylated hemoglobin (HbA1c) levels and HOMA index, and a decrease in HOMA-B index, were also observed. It was also implied that exercise training potentiates the glucose-lowering effect of the extract. Irudayaraj et al. reported that the ethyl acetate extract of *Ficus carica* Linn. showed protective effects upon glucose regulation, insulin sensitivity, and dyslipidemia [73]. An oral dose of 250 or 500 mg/kg/day for 28 days improved glucose utilization, and this was demonstrated by the increase in hexokinase and glycogen, and the decrease in glucose-6-phosphatase and fructose-1, 6-bisphosphatase. Histopathological results with improved insulin-immunostaining expression in pancreatic islets supported the cytoprotective effects of the extract. Mahmoud et al. reported that the fruit juice of *Momordica charantia* Linn., a folk medicine previously used to treat diabetes mellitus, presented antidiabetic and antioxidant effects [74]. STZ-induced diabetic male Wistar rats were post-treated with the extract for 10 mL/kg/day for 21 days, or additionally pretreated with the same dose for 14 days. The results revealed decreases in insulin resistance, serum glucose, TG, TC, serum total antioxidant capacity (TAOC), and fructosamine, while insulin levels and β-cell function improved. Antioxidant activities were demonstrated by the modulation of MDA and pancreatic GSH expression. Diaphragms isolated from the subjects displayed increases in glucose uptake when treated with 0.5% of fruit juice (0.02 mL) for 30 min. Salemi et al. described that *Morus alba* L. leaf extract and powder prevented type 2 diabetes by regulating adipokines and insulin secretion [75]. STZ-induced T2D in male Wistar rats were treated with 400 μL of *Morus alba* L. leaf extract for 6 weeks or with 25% *Morus alba* L. leaf powder as a daily diet for 6 weeks. Both samples recorded a decrease in fasting blood glucose level, insulin resistance, AST, ALT, and resistin levels. In particular, the regulation of resistin was noted as a potential mechanism of antidiabetic effects. Putakala et al. reported the beneficial effect of *Phyllanthus amarus* water extract in high-fructose diet-induced diabetic Wistar rats, enhancing insulin resistance and attenuating hepatic oxidative stress [76]. Higher plasma adiponectin, and lower fasting plasma glucose, fasting plasma insulin, HOMA, TG, TC and plasma leptin levels were observed. Histopathological impairment of the liver architecture was also prevented by *Phyllanthus amarus* water extract. Lin et al. evaluated the protective effects of *Psidium guajava* juice against metabolic disorders, as well as renal and pancreatic injury induced by type 2 diabetes [77]. Female Wistar rats, treated with high-fructose diet, nicotinamide and STZ, were administrated with a dose of 4 mL/kg for 4 weeks. The results showed reduced insulin resistance and reactive oxidative species in the renal area. Attenuation of 4-hydroxy-2-nonenal (4-HNE), IL-1β, caspase-3 expression, and LC3-B suggested that the natural product is capable of preventing autophagy, apoptosis and pyroptosis formation in the kidney and pancreas. After treatment, the size of islets improved and the arrangement of pancreatic cells recovered its regularity, also showing less hemorrhage and neutrophil gathering in the renal sections. However, it did not revert blood glucose levels in response to OGTT, and it only exhibited efficacy in HOMA-β when treated in combination with trehalose. Azmi et al. demonstrated that methanol extract from the root of *Rauwolfia serpentina* improved glucose and lipid metabolism by affecting insulin resistance or fructose absorption [78]. The administration of doses of 10, 30, and 60 mg/kg for 14 days in fructose-induced T2DM Male Wister albino mice led to changes in serum insulin, TG, HDL-C, LDL-C, VDL-c, total hemoglobin and glycosylated hemoglobin. The inhibition of 3-hydroxy-3-methylglutaryl-coenzyme A (HMG-CoA) reductase activity and the improved HMG CoA/mevalonate ratio suggested the potent efficacy of *Rauwolfia serpentine* in liver function. Ngueguim et al. reported that *Sclerocarya birrea* stem barks aqueous extract showed hypoglycemic and anti-oxidative properties against 10% of oxidized palm oil and 10% of sucrose-induced diabetic male Wistar rats [79]. The administration of 150 and 300 mg/kg doses a day for 2 weeks resulted in enhanced insulin sensitivity and decreased blood glucose levels. It was further discussed that the extract decreased the risk of glycemia, regulated dyslipidemia, and lowered blood pressure. Enhanced renal and hepatic function was observed by the decrease in ALT and AST, while the modulation of GSH, MDA, nitrite, and SOD revealed the antioxidative properties of *Sclerocarya birrea* stem. Sharma et al. elucidated that aqueous extract from *Syzygium cumini* (L.) Skeels. exhibited insulin-sensitizing, hypolipidemic, and antioxidant activity in STZ-induced T2DM male Wistar albino rats [80]. The administration of the extract with doses of 200 and 400 mg/kg/day for 21 days reduced serum glucose level, insulin resistance, lipid parameters (TC and LDL-C), and the dysfunction of β-cells in the pancreatic islet area. These effects were attributed to the increase in the activity of PPARα and PPARγ, while antioxidant effects were exhibited by the regulation of antioxidant enzymes such as SOD, catalase (CAT), glutathione peroxidase (GSH-Px), TNF-α, and thiobarbituric acid-reactive substances (TBRAS). Heras et al. suggested that the hydroethanolic extract of *Zingiber officinale* Roscoe possessed hypoglycemic, hypolipidemic, and hepatic, and metabolic-managing, effects [81]. After the administration of 250 mg/kg/day for weeks, high-fat diet-induced male Wistar rats showed decreases in plasma glucose levels, which were associated with the regulation of GLUT2 and glycerol-3-phosphate acyltransferase (GPAT). Lipid parameters (TC, HDL, and very low-density lipoprotein (VLDL)) and adiponectin levels were also reverted, which was attributed to the regulation of PPARα, PPARγ, and SREBP1. Moreover, the decrease in connective tissue growth factor (CTGF) and collagen 1 suggested the anti-fibrotic effects of ginger in the liver.

A total of thirteen studies demonstrated the antidiabetic properties of plant extracts against diabetes-induced Wistar mice. The efficacy of *Euterpe oleracea* Mart. hydroalcoholic extract was related to multiple mechanisms by regulating glucose and lipid mediators such as GLUT-4 and p-AMPK, as well as inflammatory cytokines [72]. Eight materials exhibited efficacy in restricting inflammation or oxidative stress. In particular, the increase in antioxidative enzymes such as SOD, CAT, and GSH was detected in models that were treated with *Boswellia serrata* extract [71]*, Momordica charantia* Linn. fruit juice [74], and *Sclerocarya birrea* stem barks aqueous extract [79]. *Syzygium cumini* (L.) Skeels. water extract [80], and *Zingiber officinale* Roscoe hydroethanolic extract [81] were relevant to the modulation of lipid metabolism, and both showed the upregulation of PPARα and PPARγ. Additionally, *Boswellia serrata* extract [71] reduced several complications of diabetes, by regulating AMPA and NMPA, which are known as physiological anti-depressants [82]. On the other hand, extracts from Baccharis dracunculifolia DC. Asteraceae [70], *Boswellia serrata* [71], Ficus carica Linn. [73], *Syzygium cumini* (L.) Skeels. [80], and *Zingiber officinale* Roscoe [81] were treated with more than 200 mg/kg, hence the validity of the studies should be reassessed.

#### 4.2.3. C57BL Mice

One of the most widely used laboratory mouse strains is the C57BL/6 mouse, and the commonly used substrains are C57BL/6J and C57BL/6N. The 6J substrain has been widely used in metabolic research, as it is susceptible to diet-induced obesity, type 2 diabetes, and atherosclerosis [83]. The C57BLKS/J strain inadvertently arose from the C57BL/6 background as a result of genetic contamination, but they phenotypically differ in response to diet-induced obesity and insulin resistance. C57BLKS/J mice have been characterized as relatively more resistant to diet-induced obesity compared to the C57BL/6J strain. In the severe insulin resistance state conferred by functional leptin deficiency, C57BL/6J are able to compensate for hyperglycemia with increased insulin production, whereas C57BLKS/J develops progressive hyperglycemia with islet atrophy [84]. Several plant extracts were evaluated for antidiabetic efficacy using C57BL mice (Table 4). Saito et al. described that goka fruit, also called *Acanthopanax senticosus* (Rupr. et Maxim.) Harms, alleviated obesity-associated insulin resistance and hepatic lipid accumulation in the liver [85]. High-fat diet-induced obese male C57BL/6J mice received a daily administration of 1000 mg/kg of the extract for 12 weeks, and this resulted in the downregulation of plasma glucose, liver TG, and TC. Elevated levels of phosphorylation of AMPK and expression of the cytochrome P450 7A1 (CYP7a1) gene supported the potent effects of the extract in suppressing lipid accumulation in the liver. Bae et al. suggested that the *Angelica gigas* Nakai extract exhibited ameliorating effects on hyperglycemia and hepatic steatosis through the activation of the AMP-activated protein kinase signaling pathway [86]. Male C57BL/KsJ-*db/db* mice treated with 20 or 40 mg/kg for 8 weeks showed lowered fasting glucose, TC, TG, and insulin resistance. Supplementation of this extract also increased the phosphorylation of AKt, AMPK, acetyl CoA carboxylase (ACC), and GSK3β in liver, adipose tissue, and skeletal muscles, indicating the activation of the AMPK pathway as a possible mechanism for insulin mediation. Kandouli et al. demonstrated the antidiabetic, antioxidant and anti-inflammatory properties of *Anvillea radiata* Coss. & Dur. water extract [87]. DPPH, oxygen radical absorbance capacity (ORAC) and tartrate-resistant acid phosphatase (TRAP) index were upregulated, while LDL oxidation, blood glucose and blood GSH/oxidized glutathione (GSSG) diminished. Decreased levels of HbA1C and TNF- α were also examined. *Anvillea radiata* were found to enhance body weight control capacity, reduce oxidative stress in blood, myocardial and skeletal muscles, and improve hyperlipidemic and inflammatory status. Teng et al. illustrated that the water extract of *Camellia sinensis,* also commonly called large yellow tea, enhanced metabolic syndrome and attenuated hepatic steatosis [88]. High-fat diet-induced C57BL/KsJ-db/db mice were treated with 1.5% w/w of large yellow tea for 10 weeks. Water intake, food consumption, liver-to-body weight ratio and blood glucose levels were thoroughly reduced. Furthermore, the extract was able to restore the normal hepatic structure of the mice model and reduced lipid synthesis via controlling SREBP-1 and acetyl-CoA carboxylase α. Shim et al. demonstrated that *Cichorium intybus* Linn. extract attenuated insulin and glucose metabolism in high-fat diet-induced diabetic male C57BL/6 mice [89]. The extract was treated two times a week for 50 mg/kg for 6 weeks, showing improved insulin sensitivity, glucose metabolism, and decreased blood glucose levels. The inhibition of NOD-, LRR- and pyrin domain-containing protein 3 (NLRP3) inflammasome activity led to the decrease in IL-1β expression, and regulated inducible nitric oxide synthase (iNOS), TNF-α, arginase 1 (Arg1), and IL-10. Zhao et al. reported that after the administration of *Cyclocarya paliurus* extract for 4 weeks at doses of 0.5 and 1.0 g/kg, FBG, fasting insulin level and insulin resistance index (IRI) were reduced and insulin sensitivity index (ISI) was enhanced [90]. The downregulation of serum MDA content and the upregulation of antioxidant enzymes such as SOD and GSH-Px were also observed. Chang et al. demonstrated that *Helminthostachys zeylanica* extract had the potential to relieve diabetic symptoms through lipid lowering effects [34]. In the high-fat diet-induced non-alcoholic fatty liver disease (NAFLD) C57BL/6J mice model, plasma lipid, glutamic oxaloacetic transaminase (GOT) and glutamate–pyruvate transaminase (GPT) levels decreased after the treatment at a dose of 578 mg/kg/day for 12 weeks. The alleviation of insulin resistance was also shown by the restoration of high fasting blood glucose, insulin and HOMA-IR index. Park et al. illustrated that ethanol extract of *Mori ramulus* refined insulin resistance by oxidative damage reduction and advanced glycation end-product formation [36]. A 14-week study giving 800 and 1600 mg/kg of the extract to 60% fat diet-induced C57BLKS/J-db/db mice was performed. Compared to the control group, the high-dose group showed significant decreases in blood glucose and pancreatic ROS, as well as improvements in insulin secretion, C-peptide levels in plasma and homeobox factor-1 protein expression. Yan et al. elicited the regulatory effects of *Morus alba* L. anthocyanin extract against insulin resistance by the PI3K/Akt pathway [37]. Male C57BL6/J db/db mice treated with doses of 50 and 125 mg/kg/day for 7 weeks marked an increase in adiponectin and glycogen in liver and muscles. Decreases in fasting blood glucose level, TG in the liver, TC, LDL, leptin, insulin, and insulin resistance were also noted. These results were associated with the activity of Akt, as well as the regulation of FOXO1, GSK3β, GYS2, G6pase, GSP and GSK3β. 

*Cyclocarya paliurus* extract reduced ROS production and prevented cell apoptosis and NIT-1 cell damage caused by STZ, which was intraperitoneally injected to C57BL/6J mice. All of the above results were dose-dependent. Choi et al. evaluated the capacity of *Morus alba* L. fruit extract in ameliorating insulin resistance and glucose tolerance in male C57BL/Ksj-db/db mice [91]. Daily administration of 0.5% Mulberry fruit extract for 6 weeks resulted in the downregulation of insulin resistance, blood glucose level, glucose tolerance and glycosylated hemoglobin. The supplementation of the extract significantly elevated the levels of plasma membrane GLUT4, total GLUT4, pAMPK and Akt substrate of 160 kDa (AS160), while it lowered the levels of PEPCK and G6pase in the liver. You et al. demonstrated the alleviation of 30% ethanol extract of *Nardostachys jatamansi* DC. on hyperglycemia and gluconeogenesis in the liver [92]. Body weight, blood glucose, glycosylated hemoglobin, and plasma insulin levels were significantly reduced in treating *N. jatamansi* DC. compared to the control group. Decreased HOMA-IR and OGTT index were also measured. In the liver, G6Pase and PEPCK levels were downregulated. The expressions of GLUT4, p-AS160 and p-AMPK were also suppressed in skeletal muscle. Lee et al. demonstrated that *Panax ginseng* C.A. Meyer water extract had efficacy in resolving obesity, and related disorders such as adipose inflammation, obesity, and dyslipidemia [93]. Ovariectomized female C57BL/6J mice were treated with 5% (w/w) ginseng for 15 weeks and showed the downregulation of TG, free fatty acids, and circulating insulin and glucose. Anti- inflammatory activity was detected by the increase in CD68 and TNF-α. It was also noted that angiogenic factors such as vascular endothelial growth factor A (VEGF-A) and fibroblast growth factor 2 (FGF-2), and metalloproteinase (MMP) activity, were activated, while histological examination showed the inhibition of hepatic lipid accumulation. Rozenburg et al. reported that *Sarcopoterium spinosum* extract improved insulin sensitivity in high-fat diet-induced KK-Ay mice and C57bl/6 mice [94]. In both models, *Sarcopoterium spinosum* improved glucose tolerance and insulin signaling through the phosphorylation of IR, PKB and PRAS40. The mRNA expression of proinflammatory genes, PEPCK and CD36 was reduced. However, other genes involved in carbohydrates and lipid metabolism were not affected. Liu et al. suggested a beneficial effect of *Siraitia grosvenorii* (Swingle) extract on hyperglycemia control, insulin action, glucose metabolism, dyslipidemia, and diabetic-mediated pathological changes in hepatocytes [95]. Decreased levels of G6Pase and PEPCK, and the activation of AMPK, indicated the inhibition of glucogenesis. The anti-hyperlipidemic activity of *Siraitia grosvenorii* (Swingle) extract was also demonstrated in terms of lipogenesis and fatty acid oxidation. Suppressed levels of SREBP1 and downstream lipogenic genes, including fatty acid synthase (FAS), stearoyl-CoA desaturase (SCD-1) and diacylglycerol O-acyltransferase 2 (DGAT2), regulated lipogenesis. The activation of AMPK, PPARα and CPT1a, and the inhibition of ACC, increased fatty acid oxidation. Xu et al. indicated that the total saponins from *Stauntonia chinensis* DC. had significant hypoglycemic and hypolipidemic activity in T2DM C57 db/db mice when treated at a dose of 30, 60 or 120 mg/kg for 21 days [96]. Increases in liver glycogen and HDL-C and decreases in FBG, glucose, insulin, TG and LDL-C were observed. Additionally, the treatment stimulated the phosphorylation of PI3K, Akt, AMPK and ACC, and elevated the expression of IRS-1 and GLUT4. This result is associated with the mechanism of glucose uptake and transport, and lipid metabolism. Brito-Casillas et al. demonstrated that the oral administration of *Uromastyx acanthinura* extract for 90 days, from a starting dose 0.13 g/kg, acutely reduced blood glucose level in 60% fat diet-induced type 2 diabetes C57BL/6J mice [97]. The most effective dose was 0.048 g/mouse, which showed a glucose-lowering effect 15 min after its administration. Paradoxically, long-term treatment tended to increase insulin resistance, food consumption and mean body weight. Food intake and body weight increments might explain the discrepancy between the short and long-term results and the lack of persistence of lower glucose values.

Sixteen studies with the T2DM C57BL/6 mice model were reported. *Morus alba* L. fruit extract [91], ethanol extract of *Nardostachys jatamansi* DC. [92], and *Morus alba* L. anthocyanin extract [37] demonstrated significant hypoglycemic effects. Increased levels of GLUT4 indicated improved glucose uptake and transport, while decreased levels of PEPCK and G6Pase showed the inhibition of gluconeogenesis. In lipid metabolism, *Camellia sinensis* [88], *Siraitia grosvenorii* (Swingle) extract [95] and total saponins from *Stauntonia chinensis* DC. [96] effectively suppressed fatty acid synthesis by elevating p-AMPK and inhibiting SREBP-1, ACC, and FAS. However, administrations of *Cyclocarya paliurus* extract at a dose of 0.5 or 1.0 g/kg [90], *Acanthopanax senticosus* (Rupr. et Maxim.) Harms at a dose of 1000 mg/kg [85], and *Mori ramulus* ethanol extract at a dose of 800 and 1600 mg/kg, correspond to relatively high concentrations. Thus, additional research is needed to evaluate their efficacy at a lower dose.

#### 4.2.4. KK-Ay Mice

The KK mouse is a Japanese native inbred mouse by Kondo et al. [98]. This model spontaneously exhibits type 2 diabetes-related disorders such as mild insulin resistance, hyperlipidemia and obesity, which is more severe in males [99]. Due to these characteristics, the KK-Ay mouse is a representative model for T2DM nephropathy [100]. Some studies elicited the efficacy of plant extracts on KK-Ay mice to reduce diabetic related symptoms (Table 5). Han et al. demonstrated the efficacy of *Anemarrhena asphadeloides Bge*. extract on type 2 diabetes by treating STZ induced-diabetic ICR mice models with doses of 30, 90 and 270 mg/kg for 8 weeks [27]. After treatment, glycemic control and insulin sensitivity were improved and pathological changes in pancreas, liver and kidney were attenuated. Zhang et el. elucidated that *Morella rubra* Sieb. et Zucc. fruit extract showed antidiabetic effects via insulin and glucose regulation [101]. After the supplementation for 5 weeks of 200 mg/kg/day, 1K65 diet-induced diabetic male KK-Ay mice showed downregulations in serum insulin, fasting blood glucose level, leptin, glucagon, insulin tolerance, TC, TG, LDL-C, and ALT. The hypoglycemic effects were associated with the phosphorylation of AMPK, and the regulation of PEPCK, G6Pase, and peroxisome proliferator-activated receptor gamma coactivator 1-alpha (PGC-1α). The effects on hepatic lipid metabolism were supported by decreased expressions of malic enzyme (ME), phosphatidate phosphohydrolase (PAP), acyl-CoA: cholesterol acyltransferase (ACAT), ACC1, sterol regulatory element-binding transcription factor 2 (SREBF2), and cell death-inducing DFFA-like effector A (CIDEA). Antioxidant and anti-inflammatory effects were also mentioned through the inhibition of numerous proteins such as IL-1β, as well as hepatic TNF-α, IL-6, MCP-1, plasminogen activator inhibitory (PAI-1) and lipocalin-2 (LCN-2). Wang et al. showed that *Perilla frutescens* oil treatment at a dose of 1.84 g/kg for 4 weeks decreased FBG level, and representative indicators related to T2DM, such as AST, ALT, glucose (GLU), glucose-6-phosphate dehydrogenase (G6PD), TG and TC [102]. The elevation of PI3K, p-IRS-1, p-Akt, p-AS160 and GLUT4 indicated the alleviation of insulin resistance through the PI3K/Akt signaling pathway. The relative content and diversity of intestinal microbiota were also regulated through the increased abundance of *Alloprevotella* and *Akkermansia,* and decreased amounts of *Aerococcus* and *Streptococcus.* Rozenburg et al. reported that *Sarcopoterium spinosum* extract improved insulin sensitivity in high-fat diet-induced KK-Ay mice and C57bl/6 mice [94]. In both models, *Sarcopoterium spinosum* improved glucose tolerance and insulin signaling through the phosphorylation of IR, PKB and PRAS40. MRNA expressions of proinflammatory genes, PEPCK and CD36 were reduced. However, other genes involved in carbohydrates and lipid metabolism were not affected. These results showed that some plant extracts were relevant in the inhibition of the progress of diabetes in KK-Ay mice. 

Four plant extracts were found to have significant effects on diabetes in KK-Ay mice. *Morella rubra* Sieb. et Zucc. fruit extract was effective in multiple mechanisms, as pAMPK and PEPCK were associated with the regulation of glucose, and SREBF2, PAP, and ACAT were related with the modulation of lipid metabolism [101]. Additionally, the expression of inflammatory regulators was particularly found in the liver. *Perilla frutescens* oil modulated the metabolism of glucose and its potency depended on the growth of microorganisms in the intestine [102]. *Sarcopoterium spinosum* Spach. root water extract only showed improvements in glycogen control, which was associated with the mediation of MCP-1, IκB kinase (IKK), and CD36 [94]. 

#### 4.2.5. Other Preclinical Models

Studies using other animal models except for SD rats, Wistar rats, C57BL/- mice, and KK-Ay mice were also reported to show effective antidiabetic properties (Table 6). Two studies were confirmed for ICR mice, Kunming mice, and obese (*ob/ob*) mice, respectively. For SHRSP.Z-Leprfa/IzmDmcr rats and TSOD mice, a single study was confirmed for each. 

##### ICR Mice

Institute of Cancer Research (ICR) mice have been shown to be suitable for studying metabolic syndrome. Compared with C57BL/6J mice, ICR seem to be more sensitive to the diabetogenic effects of STZ because of their difference compared to other mouse strains in terms of poly (ADP-ribose)-polymerase activities and NAD consumption in pancreatic β-cells [103]. Additionally, the combined use of high-fat diet with nicotinamide and STZ in ICR mice induced significant insulin resistance, hyperlipidemia, impaired insulin secretion, glucose intolerance, and obesity [104]. Han et al. identified the effects of *Anemarrhena asphadeloides Bge.* extract on type 2 diabetes through STZ induced-diabetic ICR mice models [27]. Mice were treated with 30, 90 and 270 mg/kg doses for 7 days, in combination with insulin, which resulted in synergistically reduced FBG levels. Tian et al. demonstrated that *Morus alba* L. water extract alleviated insulin resistance and diabetic status through the insulin signaling pathway, in which elevations of IRS1 and insulin receptor (InsR) were shown [105]. This extract neutralized the inflammation by reducing TNF-α, toll-like Receptors 2 (TLR2), and critical messengers of TLR2 signaling, such as myeloid differentiation primary-response protein 88 (MyD88), tumor necrosis factor receptor-associated factor 6 (TRAF6) and NF-κB p65. Morphological defects in the islet of the pancreas were also alleviated in terms of the number of inflammatory cells.

##### Kunming Mice

Kunming mice, the most commonly used outbred mouse line in China, possess considerable genetic variability and are a suitable outbred stock for the purposes of toxicological or pharmacological research [106]. Peng et al. demonstrated that a 50% ethanol eluent of poplar buds could control the diabetes-induced abnormalities in glucose and lipid metabolism by elevating HDL-C and reducing glucose, insulin, glycated serum protein (GSP), glycosylated hemoglobin (GHb), TC and LDL-C [107]. The activation of SOD and inhibition of MDA, IL-6, TNF-α, MCP-1 and cyclooxygenase-2 (COX-2) also indicated the mitigation of oxidative stress and inflammation, which is closely related to type 2 diabetes. The following results were derived at doses of 50 and 100 mg/kg when given to high-fat diet-, STZ-induced T2DM Kunming mice for 4 weeks. Wu et al. showed that a 30% ethanol extract of *Vernonia amygdalina* Delile (VA) suppressed hepatic gluconeogenesis via activation of AMPK. Extracts of 50, 100 and 150 mg/kg of VA were administered in high-fat diet-, STZ-induced diabetes Kunming mice for 6 weeks [108]. Fasting blood glucose, HOMA-IR and OGTT levels were dismissed. VA inhibited PEKCK and G6Pase, which are gluconeogenesis key enzymes, and vitalized AMPK activity in liver.

##### Obese (*ob/ob*) Mice 

Obese (*ob/ob*) mice are hyperphagic, hyperglycemic, and hyperinsulinemic mice models that are used for studies for obesity and diabetes [109]. Leptin-deficient *ob/ob* mice exert elevated plasma cholesterol levels, metabolic abnormalities related to the hypothalamic–pituitary–adrenal (HPA) axis and the thyroid axis [110]. These types are characterized by diabetic neuropathy and nephropathy, and thus are known to be useful study models for diabetic complications [111,112]. Koffi et al. suggested that *Cassia siamea Lam* (*Senna siamea*) (Fabaceae) ethanolic extract reduced glucose, insulin, ALT and AST levels in the leptin-deficient *ob/ob* mice model when treated for 28 days at a dose of 200 mg/kg [113]. The elevation of p-Akt and p-AMPK demonstrated the improvement of insulin and AMPK signaling in skeletal and liver tissues. It also significantly inhibited ROS production in the femoral artery, restoring the impaired endothelium-dependent relaxation to acetylcholine (Ach) in the aorta. Naowaboot et al. reported that the administration of *Vernonia cinerea* water extract (VC) at a dose of 250 or 500 mg/kg for 6 weeks in high-fat diet-induced diabetes OB mice improved insulin resistance by ameliorating glucose and lipid homeostasis [114]. At both doses, VC significantly reduced hyperglycemia, hyperinsulinemia, hyperleptinemia, and hyperlipidemia, as well as the proinflammatory cytokines. It was also shown that the treatment of VC stimulated phosphorylation of the PI3K and AMPK pathways in liver, muscle, and adipose tissue. 

##### SHRSP.Z-Leprfa/IzmDmcr Rats

The SHRSP.Z-Leprfa/IzmDmcr rats (SHRSP.ZF) strain is established by crossing stroke-prone spontaneously hypertensive rats (SHRSP) and Zucker fatty rats (ZF). Young SHRSP.ZF rats exhibit an age-dependent progression of metabolic dysfunction, which has similar characteristics to type 2 diabetes in humans [115]. Oowatari et al. elicited the therapeutic effects of *Wasabia japonica* Matsum leaf extract against adipose hypertrophy in male SHRSP/ZF rats [116]. Six weeks of daily administration of the extract with a dose of 4 g/kg decreased the amount of TG, and suppressed the expression of PPARγ as well as adipogenic genes such as ACC1, adipocyte Protein 2 (aP2), and PEPCK. It was also found that the leaf extract elevated the phosphorylation of AMPK and ACC, followed by an increase in adiponectin.

##### TSOD Mice

The TSOD (Tsumura Suzuki Obese Diabetes) mouse is a polygenic model of obese type 2 diabetes derived by selecting individual animals that developed obesity and diabetes from among ddY mice and repeating brother–sister inbreeding using the body weight and male urinary glucose level as indices [117]. Several diabetes-related phenotypes, including hypertriglyceridemia, hypercholesterolemia, impaired glucose tolerance, insulin resistance, impaired insulin secretion, and hyperglycemia, were apparently observed in TSOD mice [118]. Miki et al. investigated the preventive effects of aged garlic extract, derived from *Allium sativum* L., on TSOD mice [119]. Two percent aged garlic extract treated daily for 19 weeks reverted plasma glycated albumin levels, which was associated with the improvement of pAMPK levels. The suppression of MCP1 and FAS was only found in adipose tissues, which suggested the anti-inflammatory activities of aged garlic.

Eight studies with various experiment models, including TSOD mice, leptin-deficient *ob/ob* mice, ICR mice, Kunming mice, and SHRSP.ZF rats and patients were reported. *Cassia siamea Lam* (*Senna siamea*) (Fabaceae) ethanolic extract [113], *Morus alba* L. water extract [105] and *Vernonia cinerea* water extract [114] activated the PI3K/Akt signaling pathway, indicating the facilitation of glucose transport and metabolism. Additionally, significant anti-inflammatory and antioxidant activities were observed in *Allium sativum* L [119], *Apis mellifera* L. [120] and *Morus alba* L. water extract [105], poplar buds from *Populus* [107], and *Vernonia cinerea* water extract [114]. The upregulation of SOD and downregulation of MCP-1, CRP, TNF-α, NF-κB, MDA, IL-6 and COX were reported. These results altogether demonstrated that plant extracts have various effects on diabetic symptoms.

### 4.3. Human Studies

Two studies suggested that plant extracts had antidiabetic effects on human subjects (Table 7). Zakerkish et al. conducted a randomized double-blind study allocating 94 patients into the *Apis mellifera L*. extract group or the placebo group [120]. An intake of 1000 mg of *Apis mellifera L*. extract for 90 days improved glucose metabolism, increased HDL-C, and decreased the concentration of liver transaminase. Maintained baseline estimated glomerular filtration rate (eGFR) and reduced blood urea nitrogen (BUN) levels were observed, indicating improved renal function in T2DM patients. The extract also showed strong anti-inflammation properties by downregulating hs-CRP and TNF-α. Liu et al. conducted a randomized, double-blind, placebo-controlled study with 62 overweight or obese prediabetic patients, allocated to receive either 1.2 g/day of dietary supplement containing *Cinnamomum cassia* extract, chromium, and carnosine, or placebo [121]. A four-month treatment of dietary supplement decreased FPG, which was a primary outcome, and increased fat-free mass. However, HbA1c, insulin sensitivity markers, plasma insulin, lipids and inflammatory markers did not differ between the two groups. Whether the other components, *chromium* and carnosine, played a synergistic role also needs further examination. These results demonstrated that such extracts were effective in the relieving symptoms as well as preventing the disease progression of diabetes. However, a randomized, double-blind, placebo-controlled study with a dietary supplement containing *Cinnamomum cassia* extract, chromium and carnosine [107] needs further examination. They only manifested decreased fasting plasma glucose, while HbA1c, insulin sensitivity markers, plasma insulin, lipids and inflammatory markers were not detected. Additionally, the efficacy of another two components, chromium and carnosine, was difficult to ascertain. As such, additional examination is necessary to confirm whether the natural components played a synergistic role.

### 4.4. Clinical Trials

Two clinical studies have been conducted in Iran and Brazil, respectively, and the results demonstrated that two different leaf extracts were effective for resolving diabetic symptoms (Table 8). Rabiei et al. conducted a study to evaluate the effects of a hydroalcoholic extract of *Juglans regia* L. leaves on blood glucose level and cardiovascular risk factors in T2DM patients [122]. The trial was randomized, double-blind, placebo-controlled and parallel-grouped. Forty diabetic patients were divided into two groups; the treatment group received the capsules containing 100 mg of the extract and the control group received the capsules containing placebo, microcrystallin cellulose. Body weight, body mass index, and systolic blood pressure significantly decreased, but the alterations of blood glucose level and HOMA-IR score were not significant. Ferreira et al. implemented a trial to compare the effects of green tea extract and metformin in treating T2DM patients [123]. One hundred and twenty overweight women were randomly assigned using the double-blind method to four groups, as follows: control with cellulose, great tea group, metformin group, and the group which received capsules containing both. Green tea was found to be superior to metformin in improving glycemic control and lipid profile, such as total cholesterol and LDL-C. These results showed that plant extracts alleviated the physical conditions associated with diabetes, and were an effective cure method to diabetes, even compared to conventional drugs.

## 5. Limitation of the Studies

In the present study, the antidiabetic effects of natural substances were confirmed through various intracellular and animal experiments. One of the limitations of our research is that the research target was limited to plant extracts. Various studies have shown that phytochemicals are efficient agents in controlling diabetes via the control of glucose absorption, b-cell regeneration, insulin-releasing activity, as well as oxidation and inflammation [124,125,126]. In addition, numerous prescriptions and mixtures of natural products are known to alleviate diabetic symptoms through the modulation of gut microbiota and glucose tolerance [127,128]. However, the effect of natural compounds and mixtures on diabetes were not covered in this study. Furthermore, the research on the antidiabetic effects was limited to three mechanisms, including glucose metabolism, inflammation, and lipid metabolism. Except for these major mechanisms, there were cases that showed the regulation of other mechanisms, such as caspase-mediated apoptosis and MMP activation [71,93]. In some cases, mechanisms for mitigating the secondary complications of diabetes were identified, but an in-depth review was not conducted [54,59]. Moreover, there were some experiments wherein the distinction between mechanisms was not clear, in which one factor affected multiple treatment outcomes. In particular, since AMPK acts on both glucose and lipid metabolism, the correlation between physiological actions and therapeutic effects was indistinguishable [14]. As such, in future studies, an investigation of the causal relationships between mechanisms and effects, and the interrelation between various mechanisms, is necessary. Nevertheless, a thorough review of the antidiabetic effects of plant extracts has laid the groundwork for future studies on the therapeutic utility of plants for diabetes. 

## 6. Conclusions

The present study was conducted in the attempt to discover plant extracts that alleviated T2DM, and to explore its therapeutic mechanisms. In vitro and in vivo experiments were divided, and in vivo studies were classified based on the types of study subjects in order to clearly identify the effects of the extracts. As a result, various plant extracts were confirmed to effectively relieve insulin resistance and blood sugar levels, which were associated with the regulation of glucose metabolism, anti-inflammatory/antioxidant activities, and lipid metabolism. Although this research was limited to plant extracts and the mechanisms were only explored in three mechanisms, it identified the efficacy of plant extracts as therapeutic agents for type 2 diabetes.

## 7. Methods

Articles about T2DM in Pubmed from the National Library of Medicine and Google Scholar were collected. Extensive searching was undertaken for original articles written in English, electronically published in the most recent five years up until April 2020. Research was performed using the following keywords: “type 2 diabetes”, “insulin resistance”, and “natural product”. In vivo and in vitro studies, human studies and clinical trials were included, while bibliographies and study protocols were excluded. The study subject was limited to extracts of natural products while chemical substances and mixtures were excluded. As a result of the search, the efficacy of plant extracts was expressed via lab tests, and the mechanism-related results were classified into three categories: glucose transport and metabolism, anti-inflammation and antioxidant activity, and lipid metabolism. 

## Figures and Tables

**Figure 1 antioxidants-10-00081-f001:**
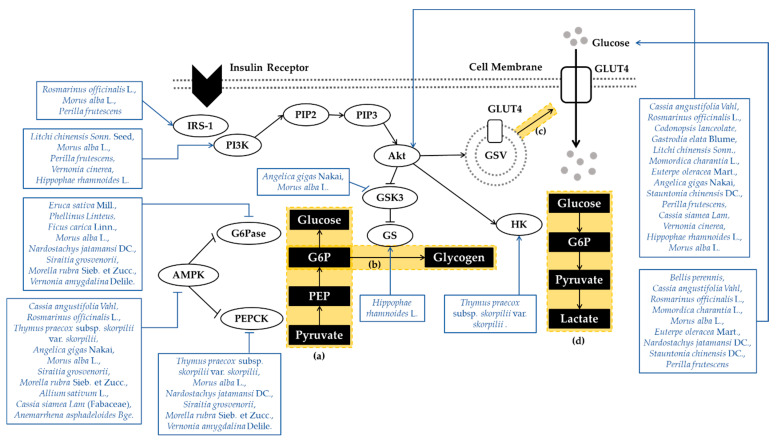
Schematic Diagram of Glucose Transport and Metabolism. Glucose transport and metabolism is associated with the activation of GLUT4 as well as various enzymes such as G6pase, PEPCK, GS, and HK. The PI3K/Akt pathway and the activity of AMPK play an essential role in glucose homeostasis. Numerous plant extracts regulated glucose metabolism in type 2 diabetes by regulating the expression of such factors. (**a**) Gluconeogenesis, (**b**) glycogen synthesis (**c**) glucose translocation, (**d**) glycolysis. (IRS-1, Insulin receptor substrate 1; PI3K, Phosphoinositide 3-kinase; PIP2, Phosphatidylinositol-4,5-bisphosphate; PIP3, Phosphatidylinositol-3,4,5-triphosphate; Akt; Protein kinase B; GSK3, glycogen synthase kinase 3; GS, glycogen synthase; G6Pase; Glucose-6-phosphatase; AMPK, Adenosine 5′-monophosphate-activated protein kinase; PEPCK, Phosphoenolpyruvate carboxykinase; GLUT4, Glucose transporter; GSV, Glut4 storage vesicle; HK, Hexokinase; G6P, Glucose 6-phosphate; PEP, Phosphoenolpyruvate).

**Figure 2 antioxidants-10-00081-f002:**
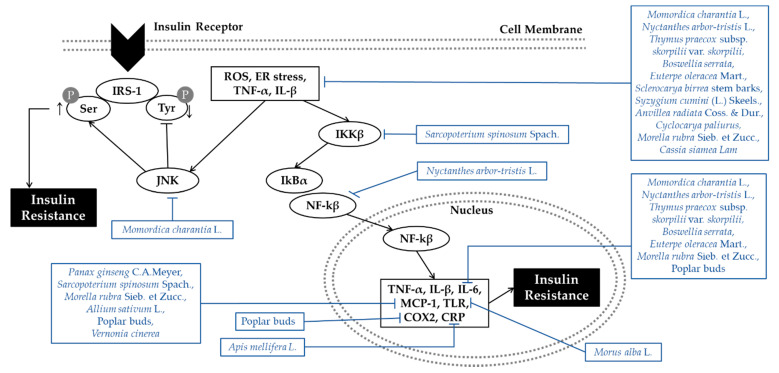
Schematic Diagram of Inflammation and Oxidative Stress. Inflammation and oxidative stress activate the JNK pathway and the IKKβ/NF-κB pathway, which leads to the increase in insulin resistance. In the nucleus, the elevation of pro-inflammatory cytokines and chemokines also promotes insulin resistance. Various plant extracts inhibited the activation of inflammatory mediators and antioxidant enzymes, which resulted in antidiabetic effects. (IRS-1, Insulin receptor substrate 1; JNK, c-Jun amino terminal kinase; IKKβ, IκB kinase-β; NF-κB, Nuclear factor kappa B; ROS, Reactive oxygen species; ER stress, Endoplasmic reticulum stress; TNF- α, Tumor necrosis factor-α; IL-1β, Interleukin-1β; IL-6, Interleukin-6; MCP-1, Monocyte chemotactic protein 1; TLR, toll-like Receptor; CRP, C-reactive protein; COX-2, Cyclooxygenase-2; Ser, serine; Tyr, Tyrosine).

**Figure 3 antioxidants-10-00081-f003:**
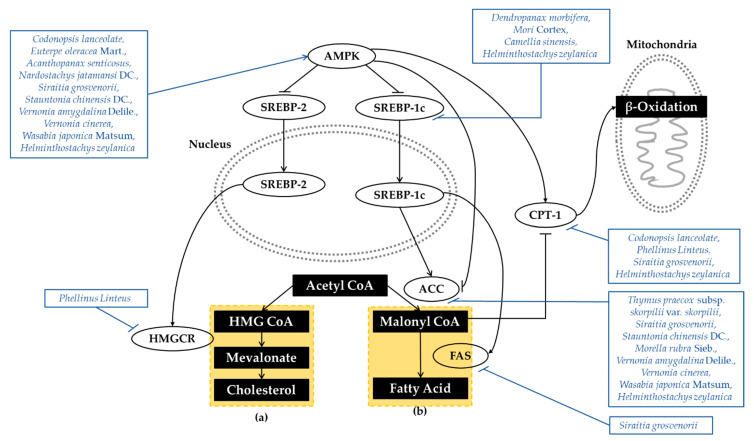
Schematic Diagram of Lipid Metabolism. AMPK inhibits SREBP and ACC, which are factors that activate cholesterol and fatty acid synthesis. AMPK is also associated with fatty acid oxidation by modulating CPT-1. Plant extracts were capable of regulating AMPK, as well as other related enzymes, which ultimately altered the state of lipid metabolism in type 2 diabetes. (**a**) Cholesterol synthesis, (**b**) fatty acid synthesis. (AMPK, Adenosine 5′-monophosphate-activated protein kinase; SREBP, Sterol regulatory element binding proteins; CPT-1, Carnitine palmitoyltransferase 1; ACC, Acetyl-CoA carboxylase; HMGCR, 3-hydroxy-3-methylglutaryl-CoA reductase; FAS, fatty acid synthase; Acetyl CoA, Acetyl-coenzyme A; HMG CoA, 3-hydroxy-3-methylglutaryl-coenzyme A).

**Table 1 antioxidants-10-00081-t001:** In vitro studies.

Source/Extract	Experimental Model	Concentration; Duration	Lab Test	Mechanisms				Reference
				Glucose Transport and Metabolism	Anti-Inflammation and Antioxidant Activity	Lipid Metabolism	Etc.	
*Anemarrhena asphadeloides Bge. extract*	3T3-L1, LKB1-deficient HeLa	30 μg/mL; 2 h		↑p-AMPK, p-ACC				[27]
*Bellis perennis* extract	CHO-K1	1 mg/L; 10 min		↑GLUT4				[28]
*Cassia angustifolia Vahl* ethanolic extract	L6	30, 60, and 120 μg/mL; 1 h		↑GLUT4, IRAP, p-AMPK, p-Akt, p-PKC			↑G protein, PLC, PKC, IP3R	[29]
*Cocoa* extract	Human primary skeletal muscle cells	10, 25 μM; 2 h	↑Basal glucose uptake					[30]
*Coptischinensis Franch* acid extract	Rin-5f	2, 10, 50, 100, 250, 500 µM; 24 h	↑GSIS ↓Insulin secretion		↑PARP-1			[31]
*Dendropanax morbifera* water extract	Mice 3T3-L1	50, 100, 300, 500 μg/mL; 7 days	↑Glucose uptake ↓Intracellular TG		↓FAS	↓PPARγ, C/EBPα, C/EBPβ, SREBP-1c,		[32]
*Eruca sativa* Mill. leaf n-haxane-soluble fraction of 95% ethanol extract	C2C12 skeletal muscle myoblast	12.5 μg/mL; 18 h	↑Glucose uptake					[33]
	H4IIE hepatoma cells	12.5 μg/mL; 16 h		↓G6Pase				
	3T3-L1 adipocyte	6.25, 12.5 μg/mL; 8 days	↑Intracellular TG					
*Helminthostachys zeylanica* extract	HuS-E/2	100 μg/mL; 18 h				↑p-AMPK, p-ACC, CPT1, PPARα, PPARδ ↓SREBP-1c, *PPARγ*		[34]
*Hippophae rhamnoides* L. fruit oil extract	IR HepG2	400 μM; 24 h	↑Glucose uptake	↑GS, PI3K, p-Akt ↓GSK-3β				[35]
*Mori ramulus* ethanol extract	INS-1	62.5, 125, 250, 500, 1000 μg/mL; 1 h	↑Insulin secretion				↑PDX-1	[36]
*Morus alba* L. anthocyanin extract	IR HepG2	50, 100, 250 μg/mL; 24 h	↑Glycogen	↑p-FOXO1, Akt2, GYS2, p-Akt, p-GSK3β ↓PGC-1α, FOXO1, G6Pase, PEPCK				[37]
*Rosmarinus officinalis* L. extract	L6	5 μg/mL; 16 h		↑GLUT4, p-Akt, p-AMPK ↓p-IRS-1, p-JNK, p-mTOR, p-p70S6K				[38]

ACC, acetyl-CoA carboxylase; GLUT4, glucose transporter 4; IRAP, insulin-regulated aminopeptidase; AMPK, AMP-activated protein kinase; IP3R, Inositol 1,4,5-trisphosphate receptor; *GSIS*, Glucose-stimulated insulin secretion; PARP-1, Poly(ADP-ribose) polymerase 1; Akt, protein kinase B; PKC, protein kinase C; SREBP-1c, Sterol response element binding protein c; PPARα, peroxisome proliferator-activated receptor α; FAS, fatty acid synthase; C/EBPα, CCAAT/Enhancer Binding Protein α; C/EBPβ, CCAAT/Enhancer Binding Protein β; G6Pase, Glucose-6-phosphatase activity; TG, Triglycerides; CPT1, Carnitine palmitoyltransferase I; SREBP-1c, sterol regulatory element-binding transcription factor 1c; HOMA-IR, homeostasis model assessment of insulin resistance; GS, glycogen synthesis; GSK-3β, glycogen synthesis kinase-3β; PI3K, Phosphoinositide 3-kinase; PDX-1, pancreatic and duodenal homeobox 1; GSK3β, glycogen synthase kinase 3β; p-Akt, Phosphorylated Akt; FOXO1, Forkhead Box O1; GYS2, glycogen synthase 2; GSP, glycated serum protein; PEPCK, phosphoenolpyruvate carboxykinase; pAMPK, phosphorylated AMP-activated protein kinase; IRS-1, insulin receptor substrate 1; JNK, c-Jun N-terminal kinase; mTOR, mammalian target of rapamycin; p70 S6K, protein S6 kinase; ↑, up-regulation; ↓, down-regulation.

**Table 2 antioxidants-10-00081-t002:** In vivo studies—SD rats.

Source/Extract	Experimental Model	Dose; Duration	Lab Test	Mechanisms				Reference
				Glucose Transport and Metabolism	Anti-Inflammation and Antioxidant Activity	Lipid Metabolism	Etc.	
*Aframomum melegueta K. Schum. fruit ethanolic extract ethyl acetate fraction*	10% fructose solution, STZ-induced type 2 diabetes SD rats	150, 300 mg/kg; 4 weeks	↑Insulin, HOMA-β, HDL-C ↓NFBG, fructosamine, HOMA-IR, TC, TG, LDL-C, AI, CRI, ALT, AST, ALP, urea, uric acid, creatinine, LDH, CK-MB	↓α -amylase, α -glucosidase				[50]
*Anemarrhena asphadeloides Bge. extract*	BCG vaccine-induced insulin resistance SD rats	20, 60, 180 mg/kg; 14 days	↑GIR					[27]
*Codonopsis lanceolate* water extract	High-fat diet-induced diabetes SD rats	0.3, 1% w/w; 8 weeks	↓Serum insulin	↑p-Akt	↓PEPCK	↑CPT-1, p- AMPK	↑SIRT-1	[51]
*Coptischinensis Franch* acid extract	High-fat diet-, STZ-induced diabetes SD rats	100 mg/kg; 8 weeks	↓Fasting blood glucose, basal insulin					[31]
*Gastrodia elata* Blume water extract	High-fat diet-induced diabetic SD rats	0.5%, 2% *Gastrodia elata* Blume water extract; 8 weeks	↑Glucose uptake ↓Serum glucose, hepatic glucose output, insulin sensitivity	↑p-Akt, pGSK-1β				[52]
*Hippophae rhamnoides* L. fruit oil extract	High-fat diet-induced type 2-diabetic SD rats	100, 200, 300 mg/kg/day; 4 weeks	↑Hepatic glycogen ↓Insulin, blood glucose, ALT, AST					[35]
*Litchi chinensis Sonn.* seeds ethanol extract	High-fat diet-, STZ-induced diabetes SD rats	30 mg/kg; 6 weeks	↓Insulin resistance, Urinary sugar, Serum ALT, Serum AST	↑PI3K, Akt, mTOR		↑FATP4		[53]
*Litchi chinensis* Sonn. seed 70% ethanol extract	High-fat diet-, STZ-induced diabetes SD rats	0.7, 1.4, 2.8 g/kg; 4 weeks	↓Blood glucose, Insulin, HOMA				↓Aβ	[54]
*Melicope lunu-ankenda* leaf extract	High-fat diet-, STZ-induced diabetes SD rats	200, 400 mg/kg; 8 weeks	↓Serum insulin, TC, TG, Serum ALT, AST					[55]
*Momordica charantia* L. 70% ethanol extract	High-fat diet-, STZ-induced diabetes SD rats	100, 200, 400 mg/kg; 8 weeks	↓Fasting serum glucose, Fasting serum insulin, HOMA-IR	↑GLUT-4, p-Akt	↓TNF-α, IL-6, JNK	↓SOCS-3	↑Akt-2, PTP-1B	[56]
*Mori Cortex* 70% alcohol extract	High-fat diet-, STZ-induced diabetes SD rats	10 g/kg; 12 weeks	↓HOMA-IR, OGTT			↓SREBP-1c, ChREBP		[57]
*Morus alba* L. leaf extract	Fructose-induced diabetic SD rats	2 g/kg/day; 4 weeks	↓Fasting blood glucose, TG, TC, LDL, HOMA-IR	↑IRS-1, PI3K, p85a, GLUT4				[58]
*Nyctanthes arbor-tristis* L. leaf ethanol extract	High-fat diet-, STZ-induced diabetes SD rats	200, 400 mg/kg; 4 weeks	↓Fasting blood glucose, Plasma insulin, VLDL, LDL, TC, TG		↓ TNF-α, IL-1β, IL-6, NF-kBp65			[59]
*Parkia biglobosa* (Jacq.) G. Don (Leguminosae) butanol extract	STZ-induced T2DM SD rats	150 mg/kg; 5 days a week, 4 weeks	↑HOMA-β, serum insulin, HDL-C, liver glycogen ↓Blood glucose level, HOMA-IR, fructosamine, ALP, urea					[60]
*Phellinus Linteus* mycelial extract	High-fat diet-, STZ-induced T2DM SD rats	300, 600 mg/kg; 8 weeks	↑Liver glycogen ↓FBG, GSP, insulin, HOMA-IR, TG, T-CHO, FFA, LDL-C, AST	↑GLUT2, GCK, ↓FBPase, G6Pase		↑ACOX1, CPT1A, LDLR ↓HMGCR		[61]
*Thymus praecox* subsp. *skorpilii* var. *skorpilii* methanolic extract	STZ/NA-induced T2DM rats	100 mg/kg; 3 weeks	↓Glucose, ALT, CR	↑AMPK, HK ↓α-glucosidase, PEPCK, SGLT-1, SGLT-2	↓TNF- α, IL-1β, IL-6	↑ACC, *PPARγ*	↑GLP-1	[62]
*Xylopiaaethiopica* (Dunal) A.Rich. fruit acetone extract	Fructose diet-induced T2DM rats	150, 300 mg/kg/day; 4 weeks	↑HOMA-β, serum insulin, ↓HOMA-IR, fructosamine, TG, TC, AI, CRI, LDL-C, ALT, liver glycogen				↓CK-MB	[63]
*Zingiber officinale* Roscoe powder	High-fat, high-fructose diet-induced prediabetic SD rats	3% ginger powder/day; 8 months	↓Insulin level, HOMA-IR, QUICK1, TG					[64]
*Ziziphus mucronata* Willd ethanol extract	STZ-induced diabetic SD rats	300 mg/kg, 5 days/week; 4 weeks	↑Serum insulin, liver glycogen ↓Blood glucose					[65]

STZ, streptozotocin; BCG, Bacillus Calmette–Guérin; GIR, glucose infusion rate; NFBG, non-fasting blood glucose; CRI, coronary risk index; LDH, lactate dehydrogenase; ALT, alanine aminotransferases; AST, aspartate aminotransferases; ALP, alkaline phosphatase; p-Akt, Phosphorylated Akt; PEPCK, phosphoenolpyruvate carboxykinase; pAMPK, phosphorylated AMP-activated protein kinase; SIRT-1, Sirtuin 1; pGSK-1β, phosphorylated glycogen synthesis kinase-1β; PI3K, Phosphoinositide 3-kinase; mTOR, Mammalian Target of Rapamycin; FATP4, Fatty acid transport protein; Aβ, Amyloid β; GLUT4, Glucose transporter 4; JNK, c-Jun N-terminal kinase; PTP-1B, protein-tyrosine phosphatase 1B; SOCS-3, Suppressor of cytokine signaling 3; SREBP-1c, Sterol response element binding protein c; ChREBP, Carbohydrate-responsive element-binding protein; IRS-1, Insulin receptor substrate 1; PI3K, Phosphoinositide 3-kinase; IL-1β, Interleukin-1β; IL-6, Interleukin-6; NF-kBp65, nuclear factor-kB p65; FBG, fasting blood glucose; GSP, glycosylated serum protein; T-CHO, total cholesterol; FFA, free fatty acids; LDL-C, low-density lipoprotein cholesterol; Bil, total bilirubin; GLUT2, glucose transporter 2; GCK, glucokinase; ACOX1, acyl-CoA oxidase 1; CPT1A, carnitine palmitoyltransferase 1A; LDLR, low-density lipoprotein receptor; FBPase, fructose-1,6-bisphosphatase; G6Pase, glucose-6-phosphatase; HMGCR, 3-hydroxy-3-methylglutaryl-CoA reductase; NA, nicotinamide; CR, creatinine; HK, Hexokinase; SGLT, sodium glucose co-transporters; TNF- α, tumor necrosis factor- α; ACC, acetyl CoA carboxylase; PPARα, Peroxisome proliferator-activated receptor α; GLP-1, glucagon-like peptide-1; HOMA-IR, insulin resistance; HOMA-B, b-cells function; TC, total cholesterol; TG, Triglycerides; AI, Atherogenic index; CK-MB, Creatine kinase; QUICKI, the quantitative insulin sensitivity check index; ↑, up-regulation; ↓, down-regulation.

**Table 3 antioxidants-10-00081-t003:** In vivo studies—Wistar rats.

Source/Extract	Experimental Model	Dose; Duration	Lab Test	Mechanisms				Reference
				Glucose Transport and Metabolism	Anti-Inflammation and Antioxidant Activity	Lipid Metabolism	Etc.	
*Baccharis dracunculifolia DC. Asteraceae* extract	MSG induced-obesity Wistar rats	400 mg/kg; 30 days	↑Insulin		↓DPPH, ABTS+			[70]
*Boswellia serrata* extract	High-fat/fructose diet-, STZ-induced type 2 diabetes Wistar rats	200, 300, 400 mg/kg; 8 weeks	↓Glucose, insulin, cholesterol, HOMA-IR		↑GSH, SOD ↓TNF- α, IL-1β, IL-6, MDA		↑AMPA, NMDA, GluR1, NR1, NR2A ↓Aβ 1-42, p-tau, caspase-3, ChE, GSK-3β	[71]
*Euterpe oleracea* Mart. hydroalcoholic extract	High-fat diet-, STZ-induced diabetes Wistar rats	200 mg/kg; 4 weeks	↑HOMA-B ↓HOMA-IR, serum leptin, serum HbA1c	↑GLUT-4, p-Akt	↓ TNF-α, IL-6	↑p-AMPK		[72]
*Ficus carica* Linn. ethyl acetate extract	STZ-induced diabetic Wistar albino rats	250, 500 mg/kg/day; 28 days	↑Glycogen ↓Plasma insulin, blood glucose, TG, TC	↑Hexokinase ↓G6Pase, fructose-1,6-bisphosphatase				[73]
*Momordica charantia* Linn. fruit juice	STZ-induced diabetic Wister rats	10 mL/kg/day; 21 days (post-treatment) or 14 days (pretreatment) and 21 days (post-treatment)	↑Insulin, LDL-C ↓Serum glucose, TG, TC, serum TAOC, fructosamine		↑GSH ↓MDA			[74]
	Diaphragms isolated from STZ-induced diabetic albino rats	0.02 mL; 30 min	↑Glucose uptake					
*Morus alba* L. leaf extract	STZ-induced T2DM Wistar rats	400 μL; 6 weeks	↓Fasting blood glucose, AST, ALT, HOMA-IR, resistin					[75]
*Morus alba* L. leaf powder	STZ-induced T2DM Wistar rats	25% of daily diet; 6 weeks						
*Phyllanthus amarus* water extract	High-fructose diet-induced Wistar rats	200 mg/kg; 60 days	↑Plasma adiponectin ↓Fasting plasma glucose, fasting plasma insulin, HOMA, TG, TC, plasma leptin		↑CAT, GPx			[76]
*Psidium guajava* juice	High-fructose diet-, NA- and STZ-induced diabetic Wistar rats	4 mL/kg; 4 weeks	↓HOMA-IR		↓H2O2, HOCl, 4-HNE, IL-1β		↓Caspase-3, LC3-B	[77]
*Rauwolfia serpentina* root methanol extract	Fructose-induced T2DM Wister albino mice	10, 30, 60 mg/kg; 14 days	↑HDL-C, Hb, HbA1c ↓Serum insulin, TG, LDL-C, VDL-c				↑HMG Co-A/Mevalonate	[78]
*Sclerocarya birrea* stem barks aqueous extract	Oxidized palm oil and sucrose-induced diabetic Wistar rats	150, 300 mg/kg/day; 2 weeks	↑Insulin sensitivity, HDL-C ↓Blood glucose, LDL-C, TG, AI, ALT, AST		↑GSH ↓MDA, SOD			[79]
*Syzygium cumini* (L.) Skeels. water extract	STZ-induced T2DM Wistar albino rats	200, 400 mg/kg/day; 21 days	↑HOMA-B , HDL-C ↓Insulin, HOMA-IR, serum glucose, serum TC, TG, LDL-C		↑SOD, CAT, GSH-Px ↓TNF-α, TBRAS	↑PPARα, PPARγ		[80]
*Zingiber officinale* Roscoe hydroethanolic extract	High-fat diet-induced Wistar rats	250 mg/kg/day; 5 weeks	↑Adiponectin ↓Insulin, plasma glucose, TC, TG, HDL, VLDL	↑GLUT2 ↓GPAT		↑PPARα, PPARγ ↓SREBP1	↓CTGF, collagen 1	[81]

MSG, monosodium glutamate; DPPH, 1,1-diphenyl-2-picrilhidrazyl; ABTS+, 2,2-azinobis-(3-ethylbenzothiazoline-6-sulfonic acid), HF, high fat; HFr, high fructose; MDA, malondialdehyde; NMDA, N-methyl-D-aspartate; GluR1, NR1, NR2A, glutamate receptors subunits; Aβ, amyloid beta; p-tau, caspase-3, ChE, cholinesterase; GSK-3β, glycogen synthase kinase-3 beta; HOMA-B, b-cells function; GLUT4, Glucose transporter 4; IL-6, Interleukin-6; p-Akt. Phosphorylated Akt; pAMPK, phosphorylated AMP-activated protein kinase; TC, total cholesterol; TG, Triglycerides; G6Pase, Glucose-6-phosphatase activity; LDL-C, Low-density lipoprotein cholesterol; TAOC, total antioxidant capacity; TNF-α, Tumor necrosis factor-α; GSH, glutathione; ALT, Alanine Transaminase; AST, Aspartate Transaminase; OGTT, oral glucose tolerance test; Gpx, Glutathione peroxidase; H2O2, Hydrogen peroxide; HOCl, Hypochlorous acid; IL-1β, Interleukin 1 beta; SOD, superoxide dismutase; HDL-C, high-density lipoprotein cholesterol; 4HNE, 4-hydroxy-2-nonenal; Hb, total hemoglobin; HbA1c, glycosylated hemoglobin; HMG-CoA, 3-hydroxy-3-methylglutaryl-coenzyme A; AI, Atherogenic index; GSH-Px, glutathione peroxidase; TBRAS, Thiobarbituric acid-reactive substances; PPARα, Peroxisome proliferator-activated receptor α; PPARγ, Peroxisome proliferator activated receptor γ; GPAT, Glycerol-3-phosphate acyltransferase; VLDL, Very low-density lipoprotein; GLUT2, Glucose transporter 2; CTGF, Connective tissue growth factor; ↑, up-regulation; ↓, down-regulation.

**Table 4 antioxidants-10-00081-t004:** In vivo studies—C57BL/- mice.

Source/Extract	Experimental Model	Dose; Duration	Lab Test	Mechanisms				Reference
				Glucose Transport and Metabolism	Anti-Inflammation and Antioxidant Activity	Lipid Metabolism	Etc.	
*Acanthopanax senticosus* (Rupr. et Maxim.) Harms fruit	High-fat diet-induced obese C57BL/6J mice	1000 mg/kg/day; 12 weeks	↓Plasma glucose, liver TG, liver TC			↑pAMPK, CYP7a1		[85]
*Angelica gigas* Nakai extract	C57BL/KsJ-db/db mice	20, 40 mg/kg; 8 weeks	↓Fasting glucose, TC, TG, HOMA-IR	↑p-AKt, pAMPK, p-ACC, p-GSK3β				[86]
*Anvillea radiata* Coss. & Dur. water extract	High-fat diet-induced C57BL/6 J mice	150 mg/kg; 12 weeks	↑DPPH, ORAC, TRAP ↓Blood glucose, blood GSH/GSSG	↓HbA1C	↓TNF-α			[87]
*Camellia sinensis* water extract	High-fat diet-induced C57BL/KsJ-db/db mice	1.5% w/w; 10 weeks	↓Serum lipid, fasting blood glucose		↓FAS	↓SREBP-1		[88]
*Cichorium intybus* Linn. extract	High-fat diet-induced diabetic C57BL/6 mice	50 mg/kg, two times a week; 6 weeks	↑Insulin sensitivity ↓Blood glucose		↑Arg1, IL-10 ↓IL-1β, iNOS, TNF-α, NLRP3			[89]
*Cyclocarya paliurus* extract	STZ-induced type 2 diabetes C57BL/6J mice	0.5, 1.0 g/kg; 4 weeks	↑ISI ↓FBG, FINS, IRI		↑SOD, GSH-Px ↓MDA, ROS			[90]
*Helminthostachys zeylanica* extract	High-fat diet-induced NAFLD C57BL/6J mice	578 mg/kg/day; 12 weeks	↑HDL-C ↓TG, TC, LDL-C, GOT, GPT, FBG, insulin, HOMA-IR					[34]
*Mori ramulus* ethanol extract	60% fat diet-induced C57BLKS/J-db/db mice	800, 1600 mg/kg; 14 weeks	↑Insulin, C-peptide ↓Fasting blood glucose		↓ ROS		↑PDX-1	[36]
*Morus alba* L. anthocyanin extract	C57BL6/J db/db mice	50, 125 mg/kg/day; 7 weeks	↑Adiponectin, glycogen ↓Blood glucose, liver TG, TC, LDL, leptin, insulin, HOMA-IR	↑p-FOXO1, p-Akt, p-GSK3β, Akt2, GYS2 ↓G6pase, GSP, GSK3β				[37]
*Morus alba* L. fruit extract	C57BL/Ksj-db/db mice	0.5% Mulberry fruit extract; 6 weeks	↑QUICKI ↓HOMA-IR, blood glucose, IPITT, IPGTT, HbA1c	↑PM-GLUT4, total GLUT4, pAMPK, AS160 ↓PEPCK, G6Pase				[91]
*Nardostachys jatamansi* DC. 30% ethanol extract	High-fat diet-induced C57BL/KsJ-db/db mice	0.2% w/w; 6 weeks	↓Fasting blood glucose, HbA1c, plasma insulin, HOMA-IR, OGTT, plasma lipid	↑GLUT4, p-AS160 ↓G6Pase, PEPCK		↑p-AMPK		[92]
*Panax ginseng* C.A.Meyer water extract	Ovariectomized C57BL/6J mice	5% (w/w) ginseng; 15 weeks	↓TG, free fatty acids, circulating insulin, glucose		↑CD68, TNFα, MCP-1		↓MMP, VEGF-A, FGF-2, MMP-2, MMP-9	[93]
*Sarcopoterium spinosum* Spach. root water extract	(1) High-fat diet-induced KK-Ay mice (2) High-fat diet-induced C57bl/6 mice	70 mg/kg; 6 weeks	↑Glycogen	↑p-GSK3β	↓MCP-1, IKK	↓CD36		[94]
*Siraitia grosvenorii* (Swingle) extract	High-fat diet, STZ-induced diabetic C57BL/6 mice	150, 300 mg/kg; 14 weeks	↑ISI, HDL-C ↓FBG, GSP, insulin, HOMA-IR, TG	↑p-AMPK ↓G6Pase, PEPCK		↑p-AMPK, p-ACC, PPARα, CPT1a ↓ACC, FAS, SREBP1, SCD-1, DGAT2		[95]
Total saponins from *Stauntonia chinensis* DC.	T2DM C57 db/db mice	30, 60, 120 mg/kg; 21 days	↑Liver glycogen, HDL-C ↓FBG, glucose, insulin, TG, LDL-C	↑IRS-1, p-PI3K, p-Akt, GLUT4		↑p-AMPK, p-ACC		[96]
*Uromastyx acanthinura* extract	60% fat diet-induced type 2 diabetes C57BL/6J mice	0.13 g/kg; 90 days	↓Glucose					[97]

TG, triglycerides; TC, total cholesterol; pAMPK, phosphorylated AMP-activated protein kinase; HOMA-IR, homeostasis model assessment of insulin resistance; p-Akt, phosphorylated Akt; ACC, acetyl CoA carboxylase; GSK-3β, glycogen synthesis kinase-3β; DPPH, 2,2-diphenyl-1-picrylhydrazyl; ORAC, oxygen radical absorbance capacity; TRAP, tartrate resistant acid phosphatase; LDL, low-density lipoproteins; GSH, glutathione; GSSG, oxidized glutathione; HbA1c, hemoglobin a1c protein; TNF-α, tumor necrosis factor–α; FAS, fatty acid synthase; SREBP-1, sterol receptor element-binding protein-1; Arg1, arginase 1; IL, interleukin; iNOS, inducible Nitric oxide synthase; ISI, insulin sensitivity index; PDX-1, pancreatic and duodenal homeobox 1FOXO1, Forkhead Box O1; GYS2, Glycogen Synthase 2; GSP, glycated serum protein; FBS, fasting blood glucose; FINS, fasting insulin level; IRI, insulin resistance index; SOD, superoxide dismutase; GSH-Px, glutathione peroxidase; MDA, malondialdehyde; ROS, reactive oxygen species; QUICKI, the quantitative insulin sensitivity check index; IPGTT, intraperitoneal glucose tolerance test; IPITT, intraperitoneal insulin tolerance test; GLUT4, glucose transporter 4; AS160, Akt substrate of 160 kDa; PEPCK, phosphoenol pyruvate carboxykinase; G6Pase, glucose-6-phosphatase activity; OGTT, oral glucose tolerance test; MCP-1, monocyte chemotactic protein 1; MMP, metalloproteinase; VEGF-A, vascular endothelial growth factor A; FGF-2, fibroblast growth factor 2; IKK, IκB kinase; CPT1a, carnitine palmitoyltransferase 1a; SCD-1, stearoyl-CoA desaturase; DGAT2, diacylglycerol O-acyltransferase 2; ↑, up-regulation; ↓, down-regulation.

**Table 5 antioxidants-10-00081-t005:** In vivo studies—KK-Ay mice.

Source/Extract	Experimental Model	Dose; Duration	Lab test	Mechanisms				Reference
				Glucose Transport and Metabolism	Anti-Inflammation and Antioxidant Activity	Lipid Metabolism	Etc.	
*Anemarrhena asphadeloides Bge. extract*	Diabetic KK-Ay mice	30, 90, 270 mg/kg; 8 weeks	↓6-h FBG, insulin, HOMA-IR					[27]
*Morella rubra* Sieb. et Zucc. fruit extract	1K65 diet-induced diabetic KK-Ay mice	200 mg/kg/day; 5 weeks	↓Serum insulin, fasting blood glucose, OGTT, ITT, ALT, TC, TG, LDL-C, leptin, glucagon	↑pAMPK ↓PEPCK, G6Pase, PGC-1α,	↓IL-1β, TNF-α, IL-6, MCP-1, PAI-1, LCN-2	↓ME, PAP, ACAT, ACC1, SREBF2, CIDEA		[101]
*Perilla frutescens* oil	High-fat/sugar diet-, STZ-induced type 2 diabetes KKAy mice	1.84 g/kg; 4 weeks	↑Insulin ↓FBG, AST, ALT, GLU, G6PD, TG, TC	↑PI3K, p-IRS-1, p-Akt, p-AS160, GLUT4			↑Alloprevotella, Akkermansia ↓Aerococcus, Streptococcus	[102]
*Sarcopoterium spinosum* Spach. root water extract	(1) High-fat diet-induced KK-Ay mice (2) High-fat diet-induced C57bl/6 mice	70 mg/kg; 6 weeks	↑Glycogen	↑p-GSK3β	↓MCP-1, IKK	↓CD36		[94]

FBG, fasting blood glucose; HOMA-IR, homeostasis model assessment of insulin resistance; OGTT, oral glucose tolerance test; ITT, Insulin tolerance test; ALT, Alanine Transaminase; TC, total cholesterol; TG, Triglycerides; LDL-C, Low-density lipoprotein cholesterol; pAMPK, phosphorylated AMP-activated protein kinase; G6Pase, Glucose-6-phosphatase activity; PGC-1α, Peroxisome proliferator-activated receptor gamma coactivator 1-alpha; PEPCK, Phosphoenol pyruvate carboxykinase; IL-1β, Interleukin-1β; TNF-α, Tumor necrosis factor-α; IL-6, Interleukin-6; MCP-1, Monocyte chemotactic protein 1; PAI-1, plasminogen activator inhibitory, LCN-2, Lipocalin-2; ME, Malic enzyme; PAP, Phosphatidate phosphohydrolase; ACAT, Acyl-CoA:cholesterol acyltransferase; ACC1, acetyl CoA carboxylase 1; SREBF2, Sterol regulatory element-binding transcription factor 2; CIDEA, Cell death-inducing DFFA-like effector A; PI3K, phosphoinositide-3 kinase; Akt, protein kinase B; IRS-1, insulin receptor substrate 1; p-AS160, phospho-Akt serine/threonine kinase; GLUT4, glucose transporter 4; AST, aspartate transaminase alanine; GLU, glucose; G6PD, glucose-6-phosphate dehydrogenase; p-GSK3β, phosphorylated-glycogen synthase kinase *3* beta; IKK, IκB kinase; CD36, cluster of differentiation 36; ↑, up-regulation; ↓, down-regulation.

**Table 6 antioxidants-10-00081-t006:** Other preclinical models.

Source/Extract	Experimental Model	Dose; Duration	Lab Test	Mechanisms				Reference
				Glucose Transport and Metabolism	Anti-Inflammation and Antioxidant Activity	Lipid Metabolism	Etc.	
*Anemarrhena asphadeloides Bge. extract*	STZ induced-diabetic ICR mice	30, 90, 270 mg/kg; 7 days	↓FBG					[27]
*Morus alba* L. water extract	High-fat/sugar diet-, STZ-induced diabetic ICR mice	2, 4, 8 g/kg; 10 weeks	↓FBG, HOMA-IR	↑IRS1, InsR	↓TNF-α, TLR2, MyD88, TRAF6, NF-κB p65			[105]
Poplar buds 50% ethanol eluent	High-fat diet-, STZ-induced type 2 diabetes Kunming mice	50, 100 mg/kg; 4 weeks	↑HDL-C ↓Glucose, insulin, GSP, GHb, TC, LDL-C		↑SOD ↓MDA, IL-6, TNF-α, MCP-1, COX-2			[107]
*Vernonia amygdalina* Delile. 30% ethanol extract	High-fat diet-, STZ-induced diabetes Kunming mice	50, 100, 150 mg/kg; 6 weeks	↓Fasting blood glucose, HOMA-IR, OGTT	↓PEPCK, G6Pase		↑p-AMPK, p-ACC		[108]
*Cassia siamea Lam* (Fabaceae) ethanolic extract	Leptin-deficient *ob/ob* mice	200 mg/kg; 28 days	↓Glucose, insulin, AST, ALT	↑p-Akt, p-AMPK,	↓ROS			[113]
*Vernonia cinerea* water extract	High-fat diet-induced diabetes OB mice	250, 500mg/kg; 6 weeks	↑Adiponectin ↓Fasting blood glucose, insulin, leptin	↑p-PI3K, p-Akt	↓TNF–α, MCP–1	↑ p-AMPK, p-ACC		[114]
*Wasabia japonica* Matsum leaf extract	SHRSP ZF rats	4 g/kg/day; 6 weeks	↑Adiponectin ↓TG			↑pAMPK, pACC ↓PPARγ, LPL, SCD1, ACC1, aP2, PEPCK		[116]
*Allium sativum* L. extract	TSOD mice	2% aged garlic extract; 19 weeks	↓Plasma glycated albumin	↑pAMPK	↓MCP1, FAS			[119]

pAMPK, phosphorylated AMP-activated protein kinase; MCP-1, monocyte chemotactic protein 1; FAS, anti-fatty acid synthase; HDL, high-density lipoprotein; eGFR, estimated glomerular filtration rate; Hb, hemoglobin; 2hpp BS, 2-h postprandial blood sugar; HOMA-IR, homeostasis model assessment of insulin resistance; HOMA-β, homeostasis model assessment of β-cell function; BUN, blood urea nitrogen; AST, aspartate transaminase alanine; ALT, alanine transaminase; hs-CRP, high sensitive C-reactive protein; TNF-α, tumor necrosis factor-α; Akt, protein kinase B; ROS, reactive oxygen species; FPG, fasting plasma glucose; FBG, fasting blood glucose; IRS1, insulin receptor substrate 1; InsR, insulin receptor; TLR2, toll-like Receptors 2; MyD88, myeloid differentiation primary-response protein 88; TRAF6, tumor necrosis factor receptor-associated factor 6; NF-κB, nuclear factor kappa B; GSP, glycated serum protein; GHb, glycosylated hemoglobin; LDL-C, low-density lipoprotein cholesterol; SOD, superoxide dismutase; MDA, malondialdehyde; IL-6, IL, interleukin-6; MCP-1, monocyte chemotactic protein 1; COX-2, cyclooxygenase-2; OGTT, oral glucose tolerance test; PEPCK, phosphoenolpyruvate carboxykinase; G6Pase, glucose 6-phosphatase; pACC, phosphorylated acetyl coenzyme A carboxylase; PI3K, phosphatidylinositol-3-kinase; TG, Triglycerides; PPARγ, peroxisome proliferator activated receptor γ; LPL, lipoprotein lipase; SCD1, stearoyl-CoA desaturase-1; ACC, acetyl-CoA carboxylase; aP2, adipocyte Protein 2; ↑, up-regulation; ↓, down-regulation.

**Table 7 antioxidants-10-00081-t007:** Human studies.

Source/Extract	Experimental Model	Dose; Duration	Lab Test	Mechanisms				Reference
				Glucose Transport and Metabolism	Anti-Inflammation and Antioxidant Activity	Lipid Metabolism	Etc.	
*Apis mellifera L*. extract	Type 2 diabetes patients	1000 mg; 90 days	↑HDL-C, eGFR ↓HbA1C, 2hpp BS, insulin, HOMA-IR, HOMA-β, BUN, AST, ALT		↓hs-CRP, TNF-α			[120]
*Cinnamomum cassia* extract, chromium, carnosine	Pre-diabetic patients	1.2 g/day; 4 months	↑Fat-free mass ↓FPG					[121]

HDL-C, high-density lipoprotein cholesterol; eGFR, estimated glomerular filtration rate; Hb, hemoglobin; 2hpp BS, 2-h postprandial blood sugar; HOMA-IR, homeostasis model assessment of insulin resistance; HOMA-β, homeostasis model assessment of β-cell function; BUN, blood urea nitrogen; AST, aspartate transaminase alanine; ALT, alanine transaminase; hs-CRP, highly sensitive C-reactive protein; TNF-α, tumor necrosis factor-α; FPG, fasting plasma glucose; ↑, up-regulation; ↓, down-regulation.

**Table 8 antioxidants-10-00081-t008:** Clinical trials.

Classification	Compound/Extract	Source	Phase	Patients	Status	Registration Number	Reference
plant	*Juglans regia* L. leaf hydroalcoholic extract	*Juglans regia* L.	Phase 2	50	Completed	IRCT138901203180 N2	[122]
plant	*Camellia sinensis* leaf 80% ethanol extract	*Camellia sinensis*	N/A	120	Completed	RBR-4bdwxs	[123]

N/A, not available.

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
