# Peer review of "Plant Extracts for Type 2 Diabetes: From Traditional Medicine to Modern Drug Discovery"

_antioxidants, 2021, doi:10.3390/antiox10010081_

Round 1

Reviewer 1 Report

This is an interesting review on plant extracts that alleviate type 2 diabetes and its therapeutic mechanisms.  Although the review is easy to read, the authors may wish to address different minor points regarding organization, clarification or corrections.

Please add the title of paragraph 4.2.5 like “other preclinical models” and better organize it.

Also, in this paragraph (as well as in table 6) authors included human studies (lines 700-706 and lines 712-718, ref 100 and 102). Please add a new paragraph, human studies before the paragraph “Clinical studies”. Suggestion: first rat, second mice and modify the conclusion.

Concerning the paragraph 4.3., you could redistribute it into the paragraph of in vitro and in vivo studies

Please review the format of abbreviations ( i.e. Akt or AKT, pAkt or p-Akt; mL or ml, caspase3, HDL, LDL VLDL,) throughout the manuscript and tables

Abbreviations as FBG FINS, ISI, NJE need to de deleted as used one time

Use T2DM and T2D throughout the manuscript and tables ( lines 324, 337, 428,448,449,459

Please modify transportation into transport throughout the manuscript (lines 97, 189, table1,

Please put in italics in vitro and in vivo throughout the manuscript

Line 109 change Activated into activated

Please specified the cells of in vitro studies ic CHO-K1 hamster ovary…, L6

Please modify dose into concentration for in vitro study (lines 196; 222,223,229,319 and in table 1)

Line 246-447 please modify into “animal model of diabetes” instead of “ in animal models that were induced with diabetes”

Line 258 please add mice after KK-Ay

Line 309 please use abbreviation LDL

Line 313 body  

Lines 337 “STZ induced T2D in male SD rats treated with…”

Line 469 “Administration of deses of 10,…”

Line 481 delete streptozotocin

Author Response

We appreciate reviewers and editors for their kind and careful comments for improving the quality of our manuscript and also sincerely hope we address our responses well to the raised comments

This is an interesting review on plant extracts that alleviate type 2 diabetes and its therapeutic mechanisms.  Although the review is easy to read, the authors may wish to address different minor points regarding organization, clarification or corrections.

Please add the title of paragraph 4.2.5 like “other preclinical models” and better organize it.

 (Response): Thanks. Revised as your comment.

Also, in this paragraph (as well as in table 6) authors included human studies (lines 700-706 and lines 712-718, ref 100 and 102). Please add a new paragraph, human studies before the paragraph “Clinical studies”. Suggestion: first rat, second mice and modify the conclusion.

 (Response): Thank you for your comments. Human studies are organized as 4.3. human studies before clinical trials. Also, 4.2.5 section is reorganized.

Concerning the paragraph 4.3., you could redistribute it into the paragraph of in vitro and in vivo studies

 (Response): Thanks. That would be better for readers. Revised.

Please review the format of abbreviations ( i.e. Akt or AKT, pAkt or p-Akt; mL or ml, caspase3, HDL, LDL VLDL,) throughout the manuscript and tables

 (Response): Thanks. The format of abbreviations were unified.

Abbreviations as FBG FINS, ISI, NJE need to de deleted as used one time

 (Response): Thanks. NJE and ISI abbreviations were removed. FINS is only in table which should be concise, so we still used the abbreviation. FBG is still used because it was used several times with NFBG.

Use T2DM and T2D throughout the manuscript and tables ( lines 324, 337, 428,448,449,459

 (Response): Thanks. Revised as T2DM.

Please modify transportation into transport throughout the manuscript (lines 97, 189, table1,

 (Response): Modified.

Please put in italics in vitro and in vivo throughout the manuscript

 (Response): Revised.

Line 109 change Activated into activated

 (Response): Revised.

Please specified the cells of in vitro studies ic CHO-K1 hamster ovary…, L6

 (Response): Revised.

Please modify dose into concentration for in vitro study (lines 196; 222,223,229,319 and in table 1)

 (Response): Revised.

Line 246-447 please modify into “animal model of diabetes” instead of “ in animal models that were induced with diabetes”

 (Response): Revised.

Line 258 please add mice after KK-Ay

 (Response): Thanks. Revised.

Line 309 please use abbreviation LDL

 (Response): Revised.

Line 313 body  

 (Response): revised.

Lines 337 “STZ induced T2D in male SD rats treated with…”

 (Response): Revised.

Line 469 “Administration of deses of 10,…”

 (Response): Revised.

Line 481 delete streptozotocin

 (Response): Thanks. Deleted.

We sincerely hope we address our responses well to the raised comments and our revised manuscript would be accepted for publication in your journal soon.

Reviewer 2 Report

The manuscript Plant Extracts for Type 2 Diabetes: From Traditional Medicine To Modern Drug Discovery fits the aim of the journal. The authors present the involvement of plant extracts in the treatment of type 2 diabetes, by synthetically presenting the results of numerous studies in vtro and in vivo. The manuscript is well systematized and written. The references are numerous, the majority of them being from the last 5 years. However, it is not very clear how the studies were selected, and I ask the authors to clarify this aspect in the manuscript.

Minor

198, 205, 752 - please also add the scientific name of the plant

Author Response

We appreciate reviewers and editors for their kind and careful comments for improving the quality of our manuscript and also sincerely hope we address our responses well to the raised comments

The manuscript Plant Extracts for Type 2 Diabetes: From Traditional Medicine To Modern Drug Discovery fits the aim of the journal. The authors present the involvement of plant extracts in the treatment of type 2 diabetes, by synthetically presenting the results of numerous studies in vtro and in vivo. The manuscript is well systematized and written. The references are numerous, the majority of them being from the last 5 years.

However, it is not very clear how the studies were selected, and I ask the authors to clarify this aspect in the manuscript.

 (Response): Thanks for your comments. “7. Methods” part is added to clarify the keywords, criteria and so forth.

Minor

198, 205, 752 - please also add the scientific name of the plant

 (Response): The scientific name for Folium Sennae is Cassia angustifolia Vahl as described in the manuscript (was line198 now line 212). The scientific names for cocoa (was line 205 not 218) and poplar buds (was line 752 not line 834) were added.

We sincerely hope we address our responses well to the raised comments and our revised manuscript would be accepted for publication in your journal soon.